# Space ionizing radiation triggers the formation of peptides and organophosphates on olivine surfaces

Ruiwen Ding[1], Shiwen Qiu[1], Xiaofan Guo[1], Min Zhang[1], Yan Liu [1] ✉, Zhiyun Zou[1], Jianxi Ying [2], Meng Zhang[3], Binquan Zhang [4] & Yufen Zhao [1,2,5]

Bioorganic molecules, such as amino acids, nucleobases, and sugars, are widely distributed in space. Here, we utilized the Chinese Space Station to carry out an investigation on the solid-state condensation reactions of these "prebiotic organic molecules" under the combined effects of ionizing radiation and forsterite. Cumulative low-dose ionizing radiation can trigger dipeptide formations and phosphorylation of riboses. Dipeptide yields increased 41-fold due to the synergistic effect of forsterite plus sodium trimetaphosphate ($P_3m$). $P_3m$ is activated upon irradiation to phosphorylate nucleosides into nucleotides. Under ionizing radiation, forsterite can promote hydroxyapatite to serve as an accessible phosphorus source for activating amino acids to form peptides. These findings indicate that complex biomolecules can be formed abiotically in space through ionizing radiation activation with the assistance of forsterite in certain radiation - resistant environments distant from planetary surfaces. It implies that apart from transporting prebiotic organic molecules to Earth, space can also provide opportunities for the in-situ assembly of ordered biomolecules from these disordered materials.

Exploring the origin of life in space and searching for extraterrestrial life are crucial for developing space resources and expanding human habitats. The extreme space environments, such as vacuum, microgravity, extreme temperatures, and radiation, pose challenges to the formation and condensation of prebiotic materials. Radiation, including ultraviolet and ionizing radiation, has been proposed as a key energy source for the early prebiotic chemical reactions on Earth[1-4]. Minerals have been proposed to play vital roles during the emergence of life[5,6], including concentrating small molecules such as amino acids[7-9], catalyzing polymerization reactions[10,11], protecting against UV radiation[12,13], selecting chiral amino acids[14,15], and participating in early genetic code events[5,16]. Most studies involving minerals for amino acid condensation into peptides have been conducted using aqueous solutions or by subjecting the samples to wetting–drying cycles[17-19]. Previous research reported that the solid–liquid interface in an electrochemical system can serve as a platform for amino acid polymerization, enabling the formation of poly-glycine films characterized by peptide bonds[20]. Some research validates that peptide synthesis can be accomplished under simulated astrophysical conditions, even in the absence of water. Among them, key pathways include the barrierless condensation of atomic carbon[21,22] and radiation-induced solid-state polymerization[23]. Notably, solid-state mineral-assisted peptide formations typically require high-temperature heating or external mechanochemical activation[24-26]. Some research focusing on peptide condensation reactions in space environment has been conducted in a simulated environment on the ground.

Analyses of samples from Murchison meteorite[27-31] and the asteroid Ryugu[32,33] revealed that the universe contains abundant

[1]Department of Chemical Biology, Fujian Key Laboratory of Chemical Biology (Xiamen University), College of Chemistry and Chemical Engineering, Xiamen University, Xiamen, Fujian, R.P. China. [2]Institute of Drug Discovery Technology, Ningbo University, Ningbo, Zhejiang, R.P. China. [3]College of Environmental Sciences and Engineering, Dalian Maritime University, Dalian, Liaoning, R.P. China. [4]National Space Science Center of the Chinese Academy of Sciences, Beijing, R.P. China. [5]Department of Chemistry, Tsinghua University, Beijing, R.P. China. ✉e-mail: stacyliu@xmu.edu.cn

"prebiotic organic molecules", including amino acids, nucleobases, and sugars. These findings demonstrate that mineral protection helps bioorganic molecules endure in space despite harsh radiation. Given the widespread distribution of these prebiotic organic molecules and an abundance of radiation energy in space, we wondered whether amino acids and nucleosides could form complex biomolecules in specific solid-state space environments with mineral assistance. Synthesizing complex biomolecules in this manner will create more opportunities for assembling ordered primitive biomolecules from disordered starting materials, both via in-situ processes and after their transport to Earth.

In this work, through the Tianzhou 6 and 7 (TZ 6 and 7) Launch Mission, we investigated the solid-state condensation and phosphorylation reactions of amino acids and nucleosides under the combined effects of forsterite and space radiation in the Space Radiobiological Exposure Facility (SREF)[34] outside the Chinese Space Station (CSS). The overall workflow is illustrated in Fig. 1. Our research aims to determine whether the combined effects of space radiation and minerals can create opportunities for forming complex life molecules under conditions where prebiotic organic molecules exist.

We selected representative bioorganic building blocks, including amino acids (Ala, Phe, Arg, and Ile) and nucleosides (A, G, C, and U), to investigate the occurrence of solid-state dehydration condensation reactions induced by space ionizing radiation with the assistance of forsterite. Forsterite, a nonclay silicate mineral, is an Mg-rich olivine mineral that is a component of Martian/Moon dust[35–37]. In space, forsterite has been found in meteorites and the dust of samples from the Stardust Mission[38]. Sodium trimetaphosphate ($P_3m$) is widely recognized as a soluble phosphorus source that can be effectively utilized in prebiotic environments[39].

In this work, $P_3m$ was used as the phosphorylating reagent to explore the solid-state phosphorylation of bioorganic molecules under the combined effects of space ionizing radiation and forsterites. In this chemical reaction model, $P_3m$ can activate amino acids to promote the dehydration condensation of amino acids to form peptides through phosphorylating amino acids. Meanwhile, $P_3m$ is capable of phosphorylating nucleosides to yield organophosphates, such as adenosine

5'-monophosphate (AMP) and adenosine 5'-triphosphate (ATP). These organophosphates constitute a crucial class of biomolecules that are of great significance for both prebiotic chemistry and modern biology.

The results show that long-term exposure to low-dose space ionizing radiation, coupled with forsterite assistance, can trigger the dehydration condensation of building blocks into complex biomolecules, including prebiotic peptides, nucleoside monophosphate (NMP), and aminoacyl nucleotides, in certain space environments that exhibit some radiation resistance. These findings improve our understanding of how the combined effects of space radiation and forsterite influence the tendency of biomolecules to undergo condensation and phosphorylation reactions. The experimental results offer different insights for identifying suitable geological environments for exploring extraterrestrial life.

## Results

### Exposure experiments onboard the CSS

The exposure samples were prepared and loaded into the experimental units according to the methods in Fig. 1 and Table 1 to simulate the main pathway by which minerals acquire organic matter and ensure uniform contact. Solutions of organic molecules were added to the mineral powder dropwise, after which the samples were freeze-dried. The space radiation experimental samples were transported to the CSS via TZ 6 (May 10, 2023), and TZ 7 (January 17, 2024), which were divided into three batches to return to Earth on October 30, 2023 (TZ 6, Unit C3), April 30, 2024 (TZ 6, Unit B3), and October 31, 2024 (TZ 7, Unit A2). Samples in Unit C3 were exposed to extravehicular cosmic radiation for 97 days, receiving a total dose of 37.75 mGy. Samples in Unit B3 were exposed to extravehicular cosmic radiation for 294 days, receiving a total dose of approximately 101.08 mGy. Samples in Unit A2 were exposed to extravehicular cosmic radiation for 174 days, receiving a total dose of approximately 53.50 mGy. The total radiation dose received by each unit was determined as the average of the measured values from pre-deployed passive detection chips, including the doses received when on standby inside the cabin (Supplementary Table 1). The extravehicular exposure experiments were maintained at a temperature of $-3\text{-}7\,°C$ under an average pressure of $0.5\text{-}0.9$

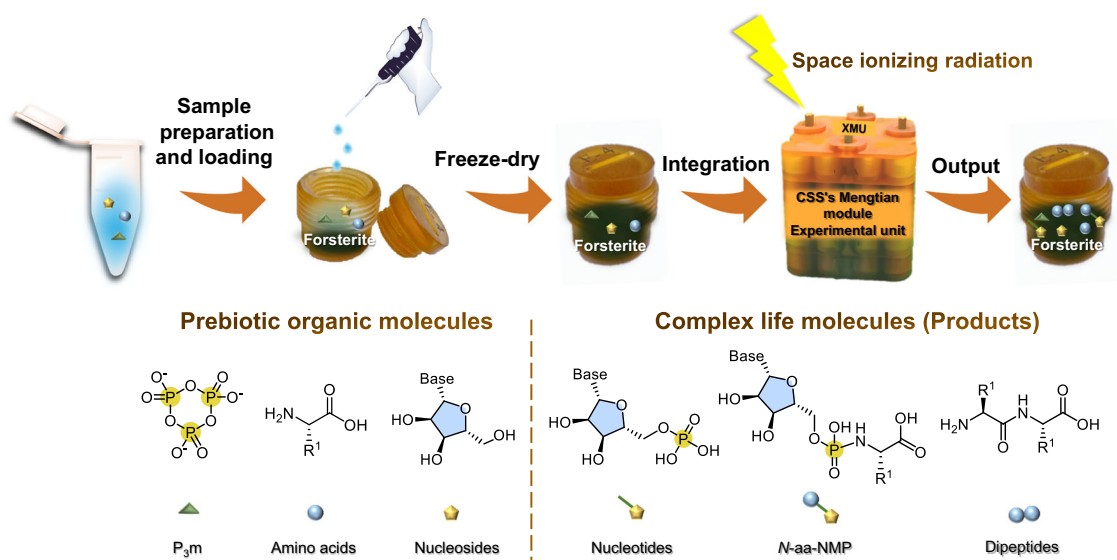

**Fig. 1 | Workflow of the space exposure experiments.** The experiments were set up in the SREF outside the Mengtian module of the CSS. To investigate how space ionizing radiation affects these reactions, both the sample container and the integrated experimental unit were fabricated from polyimide (Supplementary Fig. 1), which effectively shields against UV radiation and increases the endurance of life-related organic molecules under long-term space radiation. The samples were dropped into the minerals in solution form and then freeze-dried to complete the whole sample loading, allowing the experimental samples to be exposed to radiation in a solvent-free solid state. This sample loading method not only ensures the uniformity of organic molecules adhering to the minerals but also simulates the process by which liquid water on the star transports prebiotic organic molecules to the soil.

**Table 1 | Qualitative UPLC–HRMS analysis of the dipeptide products from the TZ 6 mission and the ground controls***

10 mM Phe + 10 mM Ala + 10 mM Arg $\xrightarrow[\text{200 μL}]{\text{Injection}}$ With/without forsterite $\xrightarrow[\text{Radiation}]{\text{Freeze-dry}}$ dipeptide

| Entry No. | Missions | Minerals | Radiation conditions | Potential dipeptide products[a] | | | | | | | | | Species number[b] |
|---|---|---|---|---|---|---|---|---|---|---|---|---|---|
| | | | | FF | AA | RR | FA | AF | FR | RF | AR | RA | |
| 1 | TZ 6 | | 97 days, 37.75 mGy | ✓ | ✓ | n.d. | ✓ | ✓ | ✓ | n.d. | n.d. | n.d. | 5 |
| 2 | Ground[c] | Forsterite | Control 1 (No radiation) | n.d. | n.d. | n.d. | n.d. | n.d. | n.d. | n.d. | n.d. | n.d. | 0 |
| 3 | | | Control 2 (17 s, 63.63 mGy) | n.d. | ✓ | n.d. | n.d. | n.d. | n.d. | n.d. | n.d. | n.d. | 1 |
| 4 | | | Control 3 (52 s, 1 Gy) | ✓ | n.d. | n.d. | n.d. | n.d. | n.d. | n.d. | n.d. | n.d. | 1 |
| 5 | | | Control 4 (120 s, 10 Gy) | ✓ | n.d. | n.d. | n.d. | n.d. | n.d. | n.d. | n.d. | n.d. | 1 |
| 6 | TZ 6 | | 97 days, 37.75 mGy | ✓ | ✓ | n.d. | ✓ | n.d. | n.d. | n.d. | n.d. | n.d. | 3 |
| 7 | Ground[c] | Blank[d] | Control 1 (No radiation) | n.d. | n.d. | n.d. | n.d. | n.d. | n.d. | n.d. | n.d. | n.d. | 0 |
| 8 | | | Control 2 (17 s, 63.63 mGy) | n.d. | n.d. | n.d. | n.d. | n.d. | n.d. | n.d. | n.d. | n.d. | 0 |
| 9 | | | Control 3 (52 s, 1 Gy) | n.d. | n.d. | n.d. | n.d. | n.d. | n.d. | n.d. | n.d. | n.d. | 0 |
| 10 | | | Control 4 (120 s, 10 Gy) | n.d. | n.d. | n.d. | n.d. | n.d. | n.d. | n.d. | n.d. | n.d. | 0 |

Notes: * Samples from TZ 6 Unit C3 were qualitatively analyzed by UPLC–HRMS (Thermo Scientific Q-Exactive ™ Plus system).

[a]FF, L-Phe-L-Phe; AA, L-Ala-L-Ala; RR, L-Arg-L-Arg; FA, L-Phe-L-Ala; AF, L-Ala-L-Phe; FR, L-Phe-L-Arg; RF, L-Arg-L-Phe; AR, L-Ala-L-Arg; RA, L-Arg-L-Ala.

[b]Number of dipeptide species detected.

[c]The ground control groups were exposed to various doses of X-ray irradiation.

[d]Blank: absence of minerals. ✓: Detected. n.d.: Not detected. All the experiments were conducted with three replicates. "s" is the abbreviation for "second". The radiation duration of the ground-control experiments provided in Table 1 is primarily intended to reflect that the experimental process involves short-term irradiation.

atmospheres (Supplementary Figs. 2–4). All treatment and characterization of forsterite are provided in the Supplementary Information (Supplementary Figs. 5–8, Supplementary Table 2).

### Space radiation combined with the assistance of forsterite triggers the dehydration condensation of amino acids into dipeptides

After the onboard experimental samples were returned to Earth, the amino acid reaction systems containing Phe, Ala, and Arg were analyzed using ultra-performance liquid chromatography–high-resolution mass spectrometry (UPLC–HRMS). The ground control experiments, in which the samples were exposed to 0 Gy, 63.63 mGy, 1 Gy, and 10 Gy of X-ray irradiation, validated the onboard findings. The analysis results are summarized in Table 1.

The reaction system included three amino acids that are capable of forming nine dipeptides. Each dipeptide product was identified by qualitative UPLC–HRMS analysis (Supplementary Figs. 9–29). Taking Phe as an example, we detected the formation of Phe-Phe (FF) dipeptide with a measurement error of $\Delta 1.92$ ppm (Supplementary Fig. 9) by extracted ion chromatogram (EIC) analysis, which was further confirmed by $MS^2$ fragmentation (Supplementary Fig. 10). No cyclic dipeptides or longer oligopeptides were detected in the reaction system.

As shown in Table 1, the TZ 6 experiment group (Entry No. 1) produced five out of nine possible dipeptide products: Phe-Phe, Ala-Ala, Phe-Ala, Ala-Phe, and Phe-Arg. Four out of the five detected dipeptides included Phe, indicating that Phe is more likely to form dipeptides under these conditions. No dipeptide products were detected in the absence of radiation (Control 1 on the ground, Entry No. 2), indicating that ionizing radiation aids in dipeptide formation. Control 2 on the ground (Entry No. 3) was exposed to low-dose radiation (63.63 mGy, 17 s), and only Phe-Phe was detected. When compared with the experimental results of Entry No. 1, the two control experimental results presented above suggest that extended exposure to low-dose radiation results in an increase in the formation of up to five dipeptide products within this reaction system. In addition, only Phe-Phe was detected in the control groups upon exposure to high radiation doses (1 Gy and 10 Gy, Entries No. 4 and 5) for a short duration. These results show that Phe, with its aromatic ring, is more likely to be activated to form dipeptides than the other two tested amino acids under the given radiation conditions.

In the TZ 6 experiment group without forsterite (Entry No. 6), three product dipeptides were detected, excluding the hetero-dipeptides Ala-Phe and Phe-Arg detected in Entry No. 1. The ability to produce dipeptides decreased and even disappeared in the mineral-deficient control groups. Forsterite exhibits radiation-resistance properties and plays a catalytic role in the peptide formation process under the given reaction conditions, particularly those of Entries No. 4 and 9.

### Impacts of exposure duration and mineral type on space radiation-induced amino acid dehydration condensation

Unit B3 was exposed to radiation outside the cabin for 294 days, with a cumulative dose of 101.08 mGy. Six types of dipeptides, including Phe-Phe, Ala-Ala, Phe-Ala, Ala-Phe, Phe-Arg (Supplementary Figs. 30–39) and Ala-Arg (Supplementary Figs. 40, 41), were detected in the same mixed amino acid systems containing forsterite by UPLC–HRMS. We analyzed the initial raw materials of the reaction system and did not observe the existence of dipeptides (Black line in Supplementary Fig. 42). This further indicated that long-term low-dose ionizing radiation can induce the formation of peptides with the assistance of forsterite.

We then replaced forsterite with montmorillonite and Martian soil simulants in the same reaction system onboard the CSS, but no dipeptides were detected. Forsterite is thus more efficient than montmorillonite and Martian soil simulants in facilitating peptide formation under the given reaction conditions.

### Quantitative analysis of dipeptides formed in the mixed amino acid reaction system under space radiation

The dipeptide products in Table 1 were quantitatively analyzed using a QTrap™ 5500 LC-MS system. In addition to enhancing diversity, forsterite increased dipeptide abundance by 2- to 14-fold compared with that of the control group (Fig. 2a). The production of Phe-Arg, which is challenging to synthesize using traditional methods, reached 29.39 pmol/50 mg forsterite, the highest among the five dipeptide-products and accounting for 65% of the total dipeptides (Fig. 2b). This indicates that forsterite plays a crucial role in promoting solid-state peptide formation, and the application of radiation in combination with forsterite can offer more strategies for obtaining complex peptides.

### Space ionizing radiation-induced dehydration condensation of nucleosides and P₃m to form NMPs

We added nucleosides and $P_3m$ to Phe, Ala, and Arg reaction systems, respectively, to examine solid-state dehydration condensation reactions with and without forsterite onboard the CSS. The experimental samples were set up in Unit C3, B3 and A2 with a cumulative radiation dose of 37.75, 101.08 and 53.50 mGy, respectively.

In addition to dipeptide products, multiple NMPs (N, nucleoside: A/G/C/U), including AMP, UMP, CMP, and GMP, were detected in the three amino acid reaction systems onboard the CSS (Supplementary Figs. 49–88). The detection amount of AMP is excellent in the three amino acid systems (Fig. 2c), which is highest among the four NMPs (Fig. 2d). The outputs of UMP, CMP, and GMP were far lower than AMP, even below the limit of quantification. For example, in the Arg system, the yield of AMP is 5 to 24 times higher than that of other NMP (Unit C3, Fig. 2d). AMP yield (306.79 pmol, Unit C3) under long-term low-dose space radiation was approximately 3 times greater than that under short-term, rapid radiation on the ground (approximately 98.14 pmol) regardless of the ground radiation dose (Fig. 2e). With the aid of forsterite, the AMP yield increased by approximately 3.8 times (Unit A2) compared to the yield without the presence of forsterite. AMP generation decreases significantly with long-term irradiation without forsterite due to AMP photodegradation (Fig. 2c, d). This also implies that forsterite possesses the function of providing radiation resistance for bioorganic molecules.

These findings indicate that nucleosides can be phosphorylated by $P_3m$ through space-ionizing radiation-activated solid-state reactions. AMP is the dominant nucleotide product and remains stable for approximately 294 days under the experimental space radiation conditions, even without mineral assistance. When compared with the EIC (Fig. 2f), the MS spectra and $MS^2$ characteristic fragmentation pattern (Supplementary Figs. 43–48) of standard 5′-NMP and 3′-NMP, the NMP produced in the reaction system was identified as 5′-NMP. Among the multiple NMPs produced, 5′-NMP is the major product, demonstrating a regioselectivity of up to 96%.

### Ground-based verification of peptides and NMPs formation stimulated by space radiation

To verify that space radiation-induced prebiotic synthesis of peptides and NMPs, solid-state reaction systems containing Phe, nucleosides (A/U/C/G), and $P_3m$ were exposed to high-dose X-ray radiation (20 Gy, 240 s) as the ground control. Three radiation experiment systems were conducted with and without forsterite to study the roles of $P_3m$ and forsterite: the Phe system, the Phe system containing $P_3m$, and the Phe system containing $P_3m$ and nucleosides (A/U/G/C). The related products of these reaction systems were analyzed via UPLC–HRMS (Supplementary Figs. 89–120) and are summarized in Supplementary Table 3.

As expected, Phe-Phe was detected in all the reaction systems. AMP and UMP were detected in the ground-based experiments only

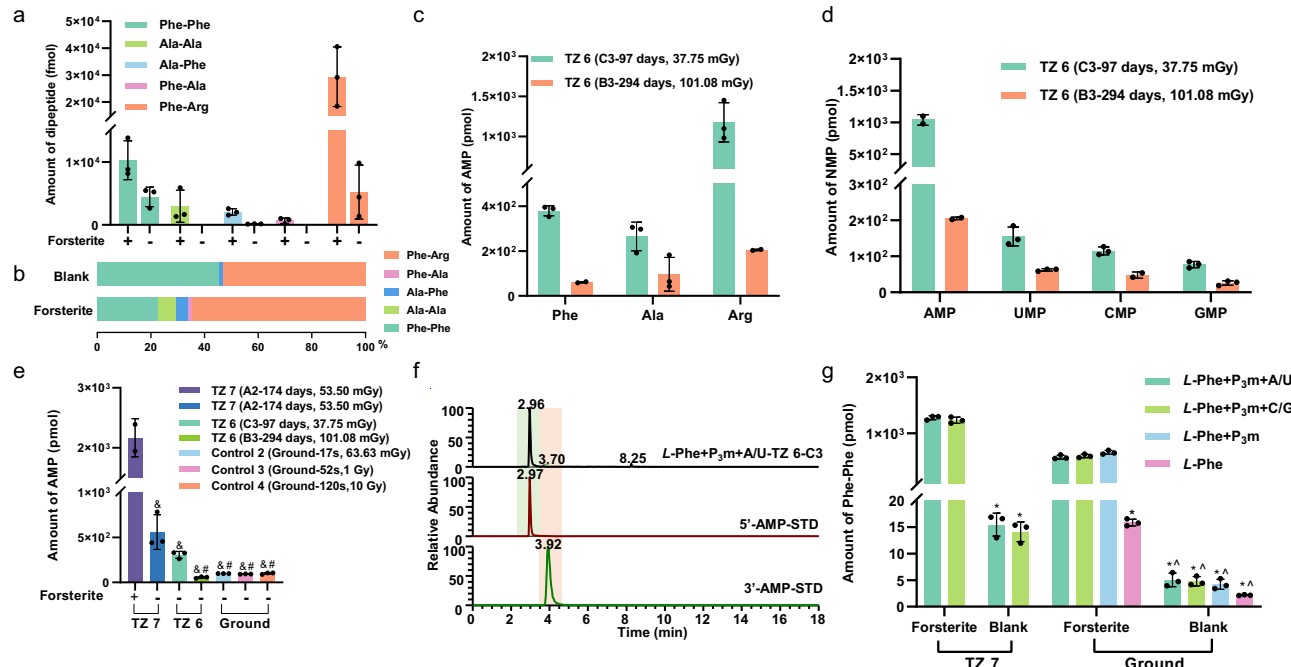

**Fig. 2 | Quantitative analysis of the products using an AB SCIEX QTrap™ 5500 LC–MS instrument. a** Amounts of product dipeptides obtained from the mixed amino acid system containing Phe, Ala, and Arg onboard the CSS. **b** Relative contents of the product dipeptides in Fig. 2a. The levels of Ala-Ala and Phe-Ala in the control group (without forsterite) were below the quantification limit. **c** Quantification analysis of AMP production in the Phe, Ala, and Arg systems containing $P_3m$ and A/U but without forsterite onboard the CSS (Unit C3 and B3). A/U stands for this reaction system contains A and U. **d** Quantification analysis of NMP (AMP, UMP, CMP, and GMP) produced in the Arg systems containing $P_3m$, and A/U or C/G. C/G stands for this reaction system contains C and G. but without forsterite onboard the CSS (Unit C3 and B3). **e** Quantification analysis of AMP production in the Phe systems containing $P_3m$ and A/U with forsterite onboard the CSS (Unit A2) and without forsterite onboard the CSS (Unit C3, B3 and A2) and the control groups exposed to varying radiation doses. $^{\&}p < 0.0001$ *vs.* the TZ 7 sample with forsterite, $^{\#}p < 0.0001$ *vs.* the TZ 6 of Unit C3 sample. **f** Extracted ion chromatogram (EIC) of the product AMP ($m/z$ 348.0704) in the reaction system of Fig. 2e set up in Unit C3 receiving a total radiation dose of 37.75 mGy onboard the CSS. 5′-AMP-STD and 3′-AMP-STD were used as standards. The peak at 2.96 min corresponds to 5′-AMP with approximately 96% regioselectivity. **g** Quantification analysis of Phe-Phe produced in the different Phe systems onboard the CSS (Unit A2) and the ground control under 20 Gy of radiation for 240 s. Blank stands for without forsterite. $^{*}p < 0.0001$ *vs.* the TZ 7 sample with forsterite, $^{\land}p < 0.0001$ *vs.* the ground sample with forsterite. Statistical analysis was conducted using GraphPad Prism version 8.0. Bars represent mean values, and error bars indicate mean ± SD (All experiments were performed with three experimental replicates. Statistical outliers were excluded from the final analysis. $n = 2 \sim 3$ independent experiments).

when forsterite was present. Since Phe-Phe and AMP were detected in both the forsterite experimental and control groups (Supplementary Table 3), they were quantitatively analyzed using a QTrap™5500 LC–MS system.

As shown in Fig. 2g, forsterite increased the Phe-Phe yield by 7- to 155-fold compared with that of the blank groups without forsterite. In the absence of forsterite, the Phe-Phe yield with $P_3m$ was 2 to 4 times greater than that without $P_3m$. In the presence of forsterite, the Phe-Phe yield with $P_3m$ increased by 35 to 41 times compared with that without $P_3m$. Thus, combining forsterite and $P_3m$ significantly enhanced the dipeptide yield.

In addition, Unit A2 experiments onboard the CSS exhibit that the Phe-Phe yield of the reaction system containing forsterite increased by 90 times than those without forsterite (Fig. 2g, Supplementary Figs. 126, 127). It indicates that forsterite indeed plays an important role in the reaction process. Furthermore, the addition of nucleosides or differences in nucleoside groups had no significant impact on dipeptide yield (Fig. 2g).

In the solid-state reaction system containing Phe, nucleosides (A/U/C/G), $P_3m$ and forsterite, in addition to dipeptides and NMPs, nucleotide amidates, 5′-aminoacyl-adenylate (5′-aa-AMP) analogs corresponding to [M + H]$^+$ at $m/z$ 495.1388 were detected (Supplementary Table 3). 5′-aa-AMPs are crucial active intermediates for peptide synthesis in all living organisms[40]. Our previous work indicated that the 5′-aa-AMP analog $N$-nucleotide amidate ($N$-aa-NMP) can be abiotically obtained in simulated hydrothermal alkaline environments[41].

Here, the EIC of the ion at $m/z$ 495.1388 showed two distinct peaks at retention times (RTs) of 5.02 and 5.29 min (Supplementary Fig. 89a), which correspond to the two isomers 2′-$N$-Phe-AMP and 5′-$N$-Phe-AMP, respectively (Supplementary Figs. 90–95). 2′-$N$-Phe-AMP exhibited instability and was susceptible to hydrolysis, resulting in the survival of only 5′-$N$-Phe-AMP after storing in the refrigerator for six months (Supplementary Fig. 89b).

## Radiation-induced prebiotic synthesis of *N*-aa-NMP occurred both onboard the CSS and on the ground

Next, to determine the structure of the $N$-aa-NMP detected in the reaction system (Supplementary Table 3, Supplementary Fig. 89), we chose Phe as the model substrate to react with adenosine and $P_3m$ with the assistance of forsterite and exposed the samples to space radiation onboard the CSS (TZ 7 Unit A2, Fig. 3a, Supplementary Fig. 125) and 20 Gy of X-ray radiation on the ground (Fig. 3b). The EIC of the $N$-Phe-AMP [M + H]$^+$ ion (exact mass $m/z$ 495.1388) determined using the optimized UPLC−HRMS method showed basic peaks at RTs of 18.83 (Figs. 3a) and 18.19 min (Fig. 3b), respectively.

The control reaction containing Phe, adenosine, and $P_3m$ was conducted in an alkaline solution (pH 11) as previously described[41] and analyzed by UPLC−HRMS/MS (Fig. 3c). The EIC of the ion at $m/z$ 495.1388 shows three distinct peaks at RTs of 11.07, 14.16 and 18.07 min, corresponding to protonated 2′-, 3′- and 5′-$N$-Phe-AMP, respectively (Fig. 3c, d). The base peaks at 18.83 min in Figs. 3a and 18.19 min in Fig. 3b differ from the base peak at 11.07 min in Fig. 3c. To

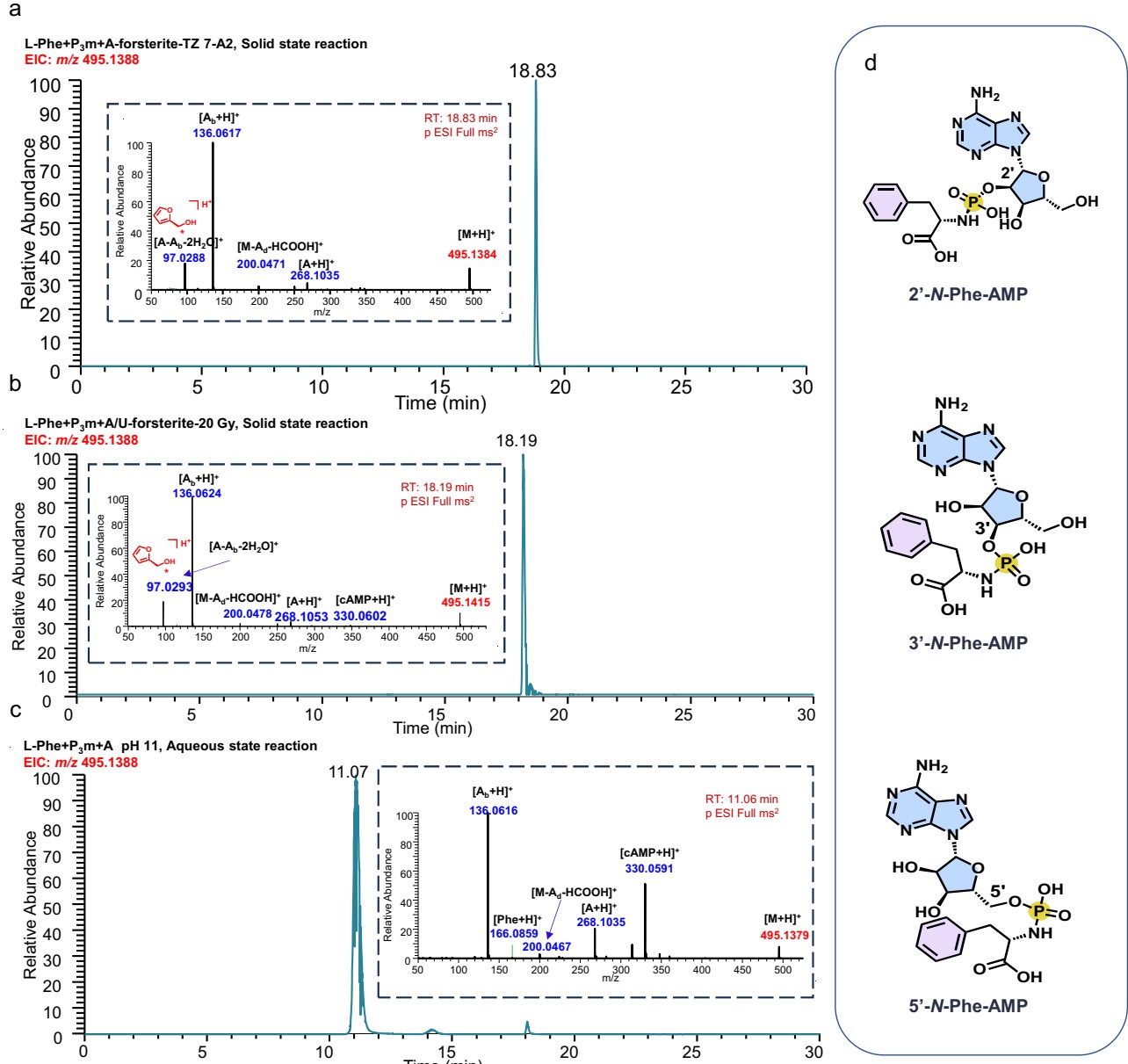

**Fig. 3 | Analysis of *N*-Phe-AMP formation in the Phe reaction systems under three distinct reaction conditions using UPLC–HRMS/MS. a** EIC of the product *N*-Phe-AMP from the forsterite solid phase reaction system exposed to space radiation onboard the CSS (TZ 7 Unit A2, 174 Days, 53.50 mGy). The insert shows the MS² spectrum of *N*-Phe-AMP (RT 18.83 min). The full scan MS spectrum and the possible fragmentation pathway have been shown as Supplementary Figs. 129, 130. **c** EIC of the product *N*-Phe-AMP produced from the alkaline aqueous solution system (pH 11) on the ground. The insert shows the MS² spectrum of 2'-*N*-Phe-AMP (RT 11.07 min). The full scan MS spectrum and the possible fragmentation pathway have been shown as Supplementary Figs. 131, 132. **d** Structures of three *N*-Phe-AMP isomers (2'-*N*-Phe-AMP, 3'-*N*-Phe-AMP and 5'-*N*-Phe-AMP). The signals in red are the molecular ion peaks, and the signals in blue are the characteristic fragment peaks. The signal marked with a red star is the characteristic fragment ion of 5'-*N*-Phe-AMP. A, Adenosine; $A_b$, adenine base; $A_d$, dehydrated adenosine; cAMP, cyclic adenylate.

compare the fragmentation patterns of the three base peaks corresponding to *N*-Phe-AMP in Fig. 3a–c, the base peaks at 18.83 min in Figs. 3a and 18.19 min in Fig. 3b both showed a characteristic fragment ion at *m/z* 97.03, which is significantly different from the pattern of the base peak at 11.07 min (Fig. 3c), indicating that the main products *N*-Phe-AMP in Fig. 3a, b are the isomer of the product in Fig. 3c.

2'-*N*-Phe-AMP was the main product in alkaline aqueous solution[41], which corresponds to the base peak at 11.07 min in Fig. 3c. The remaining two peaks at 14.16 and 18.07 min in Fig. 3c were identified as 3'-*N*-Phe-AMP and 5'-*N*-Phe-AMP[41]. By comparing the MS² fragmentation patterns with those of the standard 5'-AMP (Supplementary

Fig. 42), the base peaks at 18.83 min in Figs. 3a and 18.19 min in Fig. 3b were identified as 5'-*N*-Phe-AMP. This means that 5'-*N*-Phe-AMP is more easily produced in the ionizing radiation-triggered solid-state reaction systems both on the CSS and in the ground control group.

Comparison of Ile reaction systems triggered by ionizing radiation and alkaline aqueous solutions (Supplementary Figs. 133–140) also confirmed high regioselectivity (one peak) in solid-state producing 5'-*N*-Ile-AMP (Just one peak in Supplementary Fig. 133).

In biological systems, 5'-aa-AMPs can be synthesized via the aminoacyl-tRNA synthetase (aaRS)-catalyzed reaction of α-amino acids and ATP (Fig. 4a). Our previous research revealed that 2'-*N*-aa-AMPs,

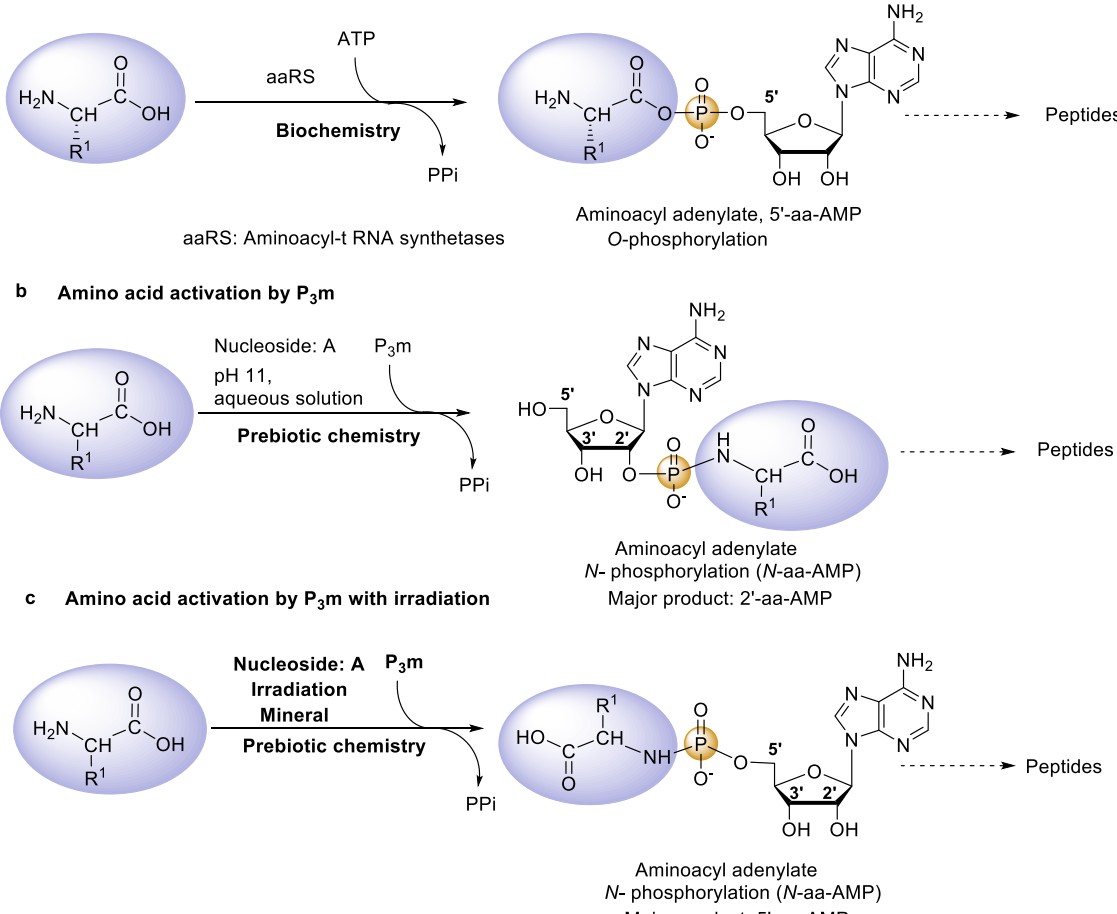

**Fig. 4 | Biotic and prebiotic pathways for aminoacyl-adenylate synthesis. a** Biotic 5′-aa-AMP synthesis pathway. **b** Prebiotic N-aa-AMP synthesis pathway in alkaline aqueous solutions. **c** Prebiotic N-aa-AMP synthesis pathway under radiation with forsterite assistance.

the analogs of 5′-aa-AMPs, can be abiotically synthesized in alkaline aqueous solutions (Fig. 4b)[41]. In this study, we found that 5′-N-Phe-AMP can be abiotically synthesized in the solid state under the combined effects of ionizing radiation and forsterite with high regioselectivity (Fig. 4c). This indicates that regardless of whether it is an aqueous-phase reaction or an anhydrous solid-phase reaction, there exist pre-biotic pathways to obtain the key intermediate aminoacyl adenylate analogs for peptide synthesis.

**Molecular mechanism of ionizing radiation-induced condensation and phosphorylation reaction**

We carried out exposure experiments in two different atmospheric environments, namely air and argon atmospheres, which were used as mutual controls. Active oxygen species, including oxygen molecules, have been reported in interstellar space[42,43]. To explore the effects of active oxygen species on our reaction systems, we designed and conducted our space exposure experiments in an air environment. The argon experiment groups were used as a control to explain the results of the dehydration condensation reaction without active oxygen spe-cies components.

We discovered that when argon was substituted for air, in addition to dipeptide and nucleoside monophosphate, more complex ami-noacyl adenylate analogs could all be detected effectively (Supple-mentary Fig. 128). That indicates that regardless of whether it is in an air or argon atmosphere, the condensation products of relevant complex life molecules can be effectively obtained.

Besides, 3 equivalents of the radical quenching agent 2, 2, 6, 6-tetramethylpiperidinooxy (TEMPO) solution were dropped into the ground control Phe reaction system. The resulting reaction systems were freeze-dried and then exposed to X-rays. The radical trapping process by TEMPO is depicted in Fig. 5a. Three parallel tests indicated that the yield of the key dipeptide product did not decrease (Fig. 5b, Supplementary Fig. 141). At the same time, the free radical products captured by TEMPO were also detected (Supplementary Figs. 142–145). It indicates that the amide and phosphoester bonds formed in the reaction system can undergo radical reactions, yet this is not the sole pathway. The related reactions predominantly proceed via an ionic reaction pathway.

The structure of forsterite consists of tetrahedral $SiO_4^{2-}$ linked by $Mg^{2+}$ cations with octahedral coordination[44], and forsterite surfaces expose both Lewis bases (Si-O⁻) and Lewis acids ($Mg^{2+}$). The electro-static interactions of these in-situ Lewis acids and bases with amino acids, nucleosides, and phosphates can facilitate the adsorption of bioorganic molecules and subsequently catalyze the progression of dehydration condensation reactions. To further explore the under-lying molecular mechanisms, we carried out a series of experiments on the ground. Upon the addition of the metal chelating agent ethylene-diaminetetraacetic acid (EDTA) to the reaction system, the yield of the dipeptide decreased by approximately twelve-fold, which confirms the significant involvement of magnesium ions (Fig. 5c). This conclusion was further corroborated by comparing different mineral substrates: the use of magnesium oxide (MgO) resulted in the highest dipeptide

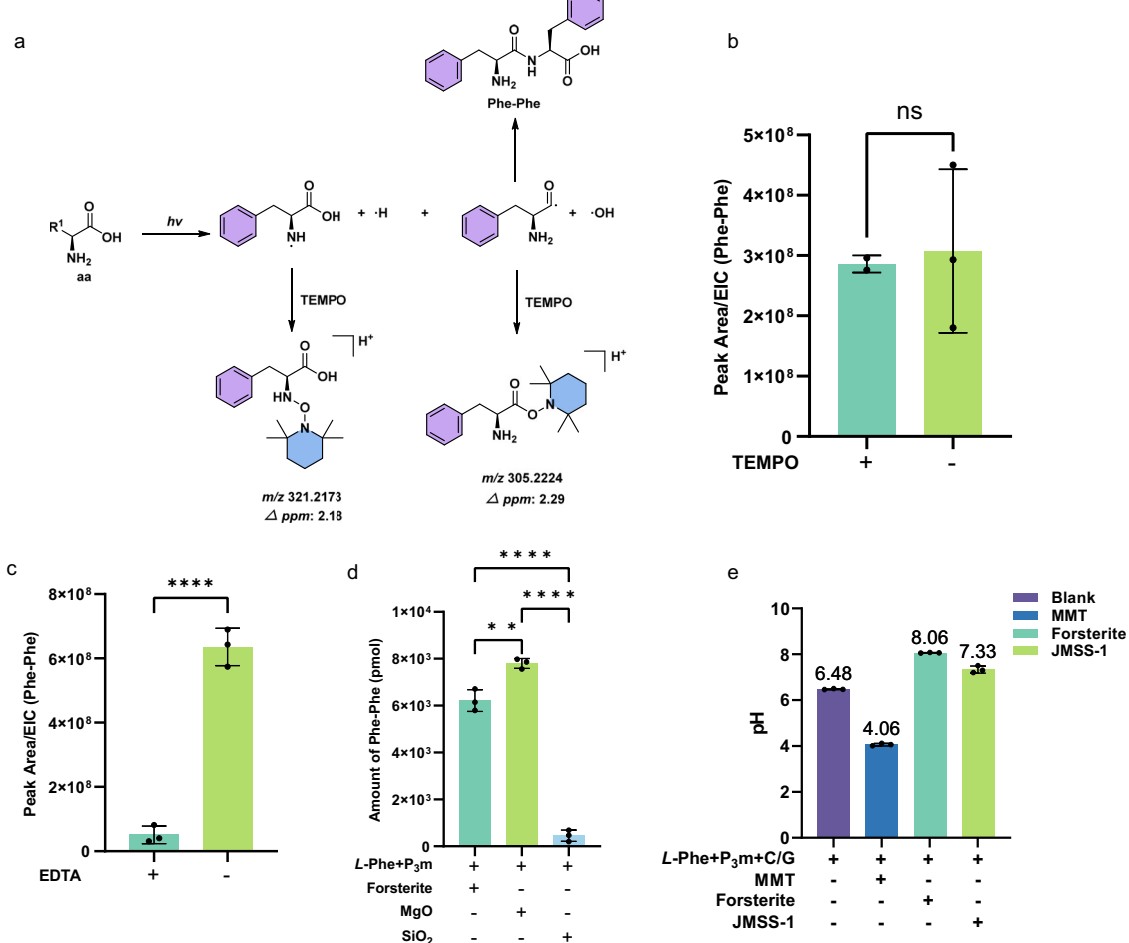

**Fig. 5 | Exploration of the mechanism of photochemical reactions. a** Trapping of reactive radical species with TEMPO, leading to the formation of acyloxyamine adducts for mechanistic elucidation. **b** The peak area analysis results of the extracted ion chromatogram (EIC) of the product Phe-Phe at $m/z$ 313.1547 from the forsterite-solid phase with/without TEMPO in the Phe reaction systems containing $P_3m$ and A/U with 63.63 mGy (17 s) X-ray radiation. **c** Quantitative analysis of Phe-Phe production with forsterite, with or without EDTA, in the Phe systems containing

$P_3m$ exposed to 63.63 mGy (17 s) X-ray radiation. **d** Quantitative analysis of Phe-Phe production with different minerals in the Phe reaction systems containing $P_3m$ under 63.63 mGy (17 s) X-ray radiation. **e** The Phe, $P_3m$ and C/G solutions at different pH values of various minerals. **: $p < 0.01$, ****: $p < 0.0001$, ns: not significant. Bars represent mean values, and error bars indicate mean ± SD (All experiments were performed with three experimental replicates. Statistical outliers were excluded from the final analysis. $n = 2 - 3$ independent experiments).

yield (7801.50 pmol), surpassing that obtained with silicon dioxide ($SiO_2$, the dipeptide yield of 454.02 pmol) or forsterite (the dipeptide yield of 6217.20 pmol, Fig. 5d**)**. It further indicates that the main catalytic component in forsterite is $Mg^{2+}$. In addition, the pH value of the reaction system containing forsterite is alkaline (pH = 8.06, Fig. 5e), which is conducive to the formation of peptide bonds and phosphoester bonds. The possible molecular mechanism of ionizing radiation-induced condensation and phosphorylation reaction under forsterite assistance is proposed in Fig. 6.

During this reaction process, $P_3m$ can be activated by ionizing radiation with the assistance of forsterite, similar to its activation in alkaline aqueous solutions. This phenomenon was also confirmed by [31]P NMR analysis of the progression of the $P_3m$-mediated ring-opening reaction under X-ray radiation (Supplementary Fig. 146).

## Discussion

Diverse life-related bioorganic molecules, minerals, and radiation energy are widely distributed across space. However, the questions of whether and how to utilize these available space resources in situ for the construction of complex biomolecules remain intriguing scientific inquiries. Typical biochemical reactions, such as the dehydration condensation and phosphorylation of life-related molecules under the

combined effects of space radiation and forsterite, were investigated in this study through a series of prebiotic chemistry experiments conducted onboard the CSS.

The experimental results indicate that long-term low-dose ionizing radiation can trigger the formation of complex biomolecules, such as dipeptides, nucleotides (NMP, NDP and NTP), and aminoacyl nucleotides, on the surface of forsterite under water-free, radiation-resistant solid-state conditions. Under the specific radiation-resistance conditions of the CSS, the Phe-Phe dipeptide exhibited remarkable stability, showing no detectable degradation. The photochemical stability of different chiral configuration Phe-Phe dipeptides ($L$-Phe-$L$-Phe, $L$-Phe-$D$-Phe and $D$-Phe-$D$-Phe) doesn't exhibit a significant difference (Supplementary Fig. 169) with a total dose of 37.75 mGy without minerals in the C3 Unit. When compared with the other tested amino acids, Phe exhibits a higher sensitivity and is more prone to form peptides. Nevertheless, the increase in radiation duration and dose led to the degradation of the generated dipeptides.

$P_3m$ can be activated by ionizing radiation to participate in peptide formation and the phosphorylation of nucleosides to obtain nucleotides. Combining $P_3m$ with forsterite increased dipeptide formation by up to 41-fold and the diversity of monophosphate nucleotides. 2', 3' and 5'-AMP can all be effectively produced without obvious

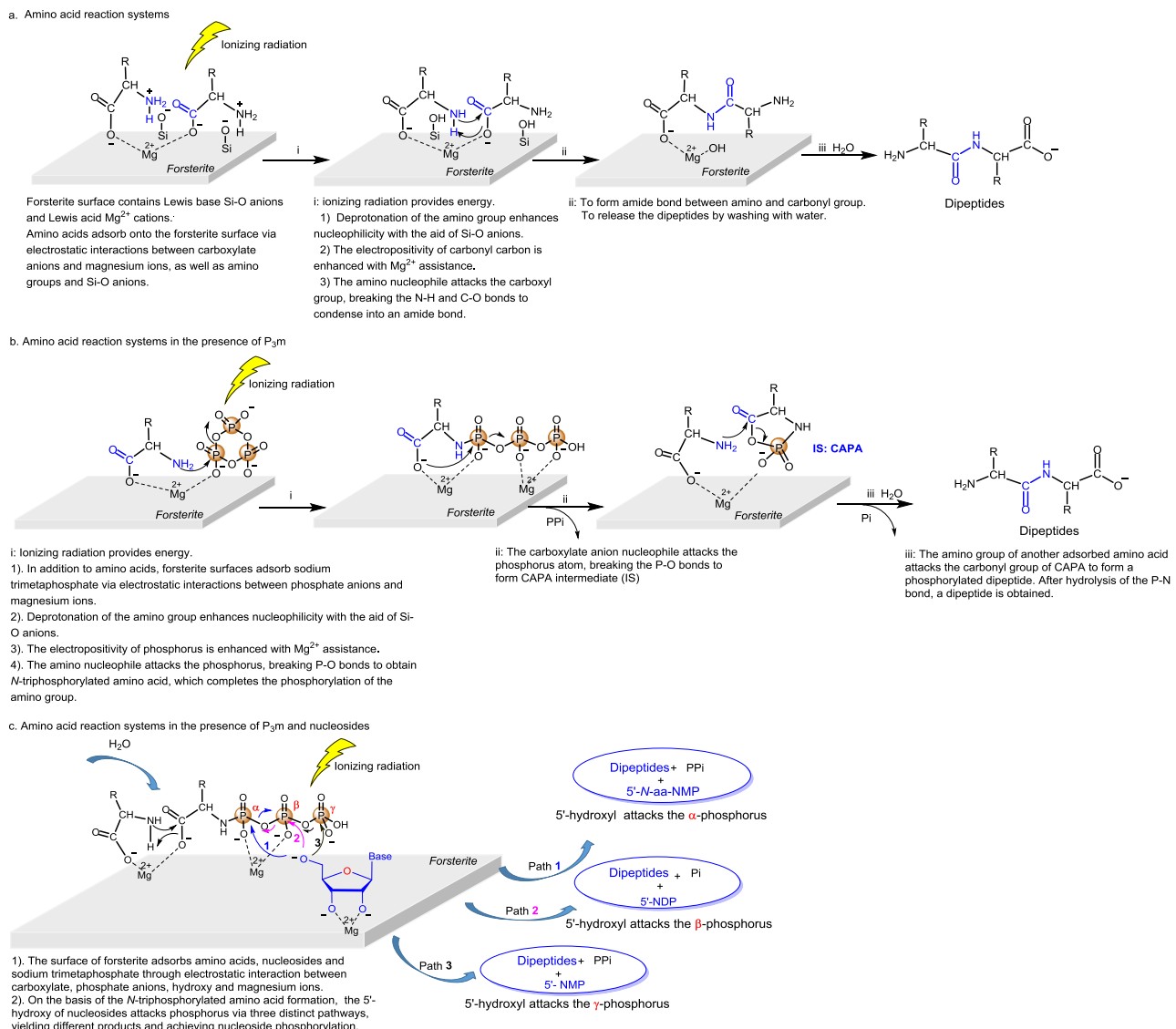

**Fig. 6 | Possible molecular mechanism of ionizing radiation-induced amino acid dehydration condensation and phosphorylation facilitated by forsterite.** In addition to dipeptide products, 5′-N-aa-NMP (Unit A2 TZ 7 and the ground control), 5′-NDP (Unit A2 TZ 7, Supplementary Figs. 121, 122) and 5′-NMP (Unit A2 TZ 7 and Unit C3, B3 TZ 6, Supplementary Figs. 123, 124) were also detected. Path 1 is drawn with blue arrows, Path 2 is drawn with red arrows and Path 3 is drawn with black arrows.

preference. 5′-AMP is an important structural unit of organic cofactors, such as ATP, FAD, NAD(H), NADP(H) and coenzyme A, which may be vestiges of ribozymes[45,46]. The amount of 5′-AMP produced increased threefold under long-term low-dose radiation exposure in space compared with that after short-term exposure on the ground.

Apatite is the primary form of phosphorus resources, not only on Earth but also present across space, such as Mars[47]. However, apatite is insoluble in water, a characteristic that limits its application in prebiotic chemistry. In this work, after replacing P₃m with hydroxyapatite in our reaction system on the ground, we observed that hydroxyapatite can be utilized to promote peptide formation under the combined effects of ionizing radiation and forsterite (Supplementary Figs.147–159). These experimental results offer an approach to addressing the "phosphorus limitation" in the origin of life.

Recent research has reported that heat flows can solubilize apatite, subsequently producing P₃m and then enhancing phosphate availability for prebiotic chemistry[48]. In 2023, Frank Postberg et al. reported the detection of huge phosphates originating from Enceladus's ocean by Cassini's Cosmic Dust Analyzer (CDA)[49]. In addition, the suitable temperature[50] and a similar heat flow system[51] with the above

research report have been found and confirmed in the Enceladus plume. Therefore, we speculate that P₃m may exist across space, such as in Enceladus's ocean. Due to the high reactivity of P₃m in an alkaline aqueous environment, P₃m has difficulty surviving in alkaline aqueous solutions unless it is in an anhydrous solid state. It is prone to hydrolyze into orthophosphate, leading to a concentration too low to be detectable. That may be one of the reasons why P₃m has not been detected in that mission by CDA.

Finally, these discoveries indicate that complex biomolecules can be generated utilizing space resources, particularly with the combined effects of olivine minerals and space radiation. Geological environments rich in forsterite deep below the surface, exposed to prolonged low-dose ionization radiation, can foster the emergence of complex biomolecules in situ. Apart from the radiation resistance effect, which is a common function of minerals, we propose that the characteristic functions of forsterite in our reaction system primarily encompass the following three aspects: 1) to offer an alkaline environment, which facilitates the formation of peptide bonds and phosphoester bonds; 2) adsorption induced by electrostatic interaction; 3) the catalytic effect centered around magnesium ions.

Our current research offers diverse perspectives and experimental references for identifying geological environments appropriate for the exploration of extraterrestrial life. Additionally, our findings also imply that using milder pretreatment methods (such as 20 mM $NH_4HCO_3$, pH = 8.86) may allow the detection of more complex biomolecules from extraterrestrial samples such as meteorites and interstellar dust.

## Methods
### Materials
$L$-phenylalanine (Phe), $L$-alanine (Ala), $L$-arginine (Arg), L-Isoleucine (Ile), $D$-adenosine (A), $D$-guanosine (G), $D$-uridine (U), $D$-cytidine (C) were purchased from Energy Chemical Ltd. (Shanghai, China). Unless otherwise specified in this study, all the amino acids were of the $L$-configuration, and the nucleosides were of the $D$-configuration. Sodium trimetaphosphate ($P_3m$) was purchased from Aladdin Chemical Ltd. (Shanghai, China). Acetonitrile and methanol (high-performance liquid chromatography grade) were purchased from Spectrum Chemical Manufacturing Corp. (Shanghai, China). Ultrapure water with a resistivity of 18.2 MΩ·cm was obtained from our lab's Milli-Q Pure Water System (Millipore, Bedford, MA).

Forsterite was purchased from Anhui SCIEX Technology Co., Ltd. Montmorillonite was purchased from Aladdin Industrial Corporation (Shanghai, China). Martian soil simulant was from the Lunar and Planetary Science Research Center, Institute of Geochemistry, Chinese Academy of Sciences. The minerals are cleaned up and characterized before use. The detailed methods will be found in the Supplementary Information.

### Exposure experiments onboard the China Space Station
All exposure samples were loaded into an independent sample container, sealed with internal and external double-threaded caps, and securely placed on sample fixing disks with external threads (Supplementary Fig. 1). Five sample disks were loaded with 80 independent samples to form an independent experiment unit. All experiments were conducted with three replicates within the facility's designated temperature and pressure range without specific controls.

The sample unit was placed in the Space Radiobiological Exposure Facility (SREF), which was designed and developed by the National Space Science Center of the Chinese Academy of Sciences and Dalian Maritime University. A comprehensive characterization of SREF outside the China Space Station has been documented in published literature[34]. The total radiation dose received by each unit was monitored from pre-deployed lithium 6 or 7 passive detection chips in the sample unit, including the doses received during standby time inside the cabin. All the values presented in this work are based on the calculated average dose to ensure a consistent frame of reference. The radiation absorbed in this experiment unit mainly consisted of radiation with LET values less than 10 keV/μm, including protons, neutrons and some heavy ion radiation. More detailed information is illustrated in the Supplementary Information, including the estimation of the radiation yield ($G$ value).

All onboard experimental results were simultaneously verified by ground control experiments, which were exposed to the corresponding dose of X-ray irradiation.

### Preparation and loading of experimental samples
The solutions of $P_3m$, amino acids, and nucleosides with the same 30 mM concentration were thoroughly mixed in equal volumes in a 1.5 mL EP tube.

To aliquot 400 μL of the aforementioned mixed solution into two polyimide sample containers. Among the above two sample containers, one containing forsterite was selected as the experimental group, while the other one without forsterite was used as the blank control group.

More detailed information is illustrated in the Supplementary Information.

### HPLC-HRMS analysis
All the sample systems were analyzed using reverse-phase liquid chromatography high-resolution mass spectrometry (Thermo Ultimate 3000 HPLC and Thermo Q Exactive, Thermofisher, USA).

The quantitative analysis of the dipeptide was performed using QTrap™ 5500 (AB SCIEX). The detailed analysis methods are illustrated in the Supplementary Information.

### Statistical analysis
Data are expressed as mean ± SD. The two-way ANOVA test tested the normality of the data distribution; a $p$-value greater than 0.05 indicates that the observed distribution of a variable is not statistically different from the normal distribution. Data analysis was done with GraphPad Prism software version 8.

## Data availability
All the data that support the findings of this study, including all the HPLC, Extraction Ion Chromatography, MS$^2$ spectra, etc., are presented in the manuscript and Supplementary Information or are available from the corresponding author upon request. Source data are provided with this paper.

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

## Acknowledgments

This work was financially supported by the Space Application System of China Manned Space Program (KJZ-YY-WSM01, Y. L.; YYWT-0901-EXP-16, Y. F. Z., Y. L; KJZ-YY-NSM0406,Y. F. Z, Y. L), the National Natural Science Foundation of China (No. 92256203, Y. F. Z, Y. L; No. 42388101, Y. F. Z) and the Ningbo Top Talent Project (No. 215-432094250, Y. F. Z.). Thanks to Professor Huijun Zhang at Xiamen University for the helpful discussion and technical support of the glove box. Thanks to Professor Gaosen Zhang at the Northwest Institute of Eco-Environment and Resources, Chinese Academy of Sciences, for many constructive discussions on radiation chemistry. Thanks to Professor Jihua Hao at the University of Science and Technology of China for many constructive discussions on the phosphorus source. Thanks to Mr. Gengqi Wu, an undergraduate student at Xiamen University, for his meticulous literature retrieval and sorting work during the manuscript revision period.

## Author contributions

R. W. Ding performed the experiments and wrote the draft. S.W. Qiu and Z.Y. Zou repeated part of the experiments. X.F. Guo participated in analyzing data and drawing Figures. Min. Zhang conducted a $^{31}$P NMR experiment. Y. Liu designed, guided and supervised this work. Y. Liu also participated in writing the manuscript. J. X. Ying provided guidance on

experimental techniques. Meng. Zhang and B.Q. Zhang provided technical support for the on-orbit experiment facility and the radiation environment dose parameters. Y.F. Zhao guided this work and provided support for experimental conditions.

## Competing interests

The authors declare no competing interests.
