## [Transparent Peer Review file · Nature Communications]

Space ionizing radiation triggers the formation of peptides and organophosphates on olivine surfaces

Corresponding Author: Professor YAN Liu

Version 0:

Reviewer comments:

Reviewer #1

(Remarks to the Author)

Review of the Article: "Space ionizing radiation triggers the formation of peptides and organophosphates on olivine surfaces"

This article investigates the formation of dipeptides and the phosphorylation of biomolecules in the solid state under exposure to space radiation in Earth's orbit. It also examines the role of different silicates in influencing these chemical processes.

The study raises two major considerations.

First, the polymerization or condensation of amino acids under energetic processing has been reported in numerous earlier studies, which the introduction does not adequately review. Likewise, the catalytic role of silicates in prebiotic chemistry has been addressed repeatedly in the literature. Moreover, the reactant molecules used in this work are unlikely to exist beyond Earth in concentrations sufficient to support the reported reactions.

Nevertheless, the study has a notable strength: a portion of the experiments was conducted aboard the Space Station, under genuine open-space conditions. This lends the results unique value and broad appeal. With revisions, the article could merit publication.

Major Issues

1. Experiments performed in space

The exposure experiments conducted onboard the space station require a more detailed description.

- o What types of radiation reached the samples, and which are expected to play the dominant role in the studied processes?
- o How might other radiation types have contributed?
- o Were all samples exposed to the same dose of radiation, or could their positioning have caused shielding effects?
- o What atmosphere was present above the samples, why was it chosen, and what natural conditions does it aim to replicate?

2. Literature review

The introduction should more thoroughly review prior research on peptide formation, particularly in the solid state (e.g., <https://doi.org/10.1038/ncomms9385>, <https://doi.org/10.1039/C9RA00179D>) and in astrophysical environments (e.g., <https://doi.org/10.21203/rs.3.rs-6939044/v1>, <https://doi.org/10.1126/sciadv.adj7179>, <https://doi.org/10.1038/s41550-021-01577-9>, <https://doi.org/10.1088/0004-637x/765/2/111>).

3. Stability of biomolecules under radiation

The potentially most compelling aspect of these experiments — the stability of biomolecules under ionizing radiation - is not addressed. Did the authors assess the degradation of dipeptides or other molecules? This information would be valuable in itself and especially relevant for interpreting differences in molecular abundance across samples.

Minor Issues

- Line 21: Define AMP.
- Lines 21–22: Clarify the phrase “Hydroxyapatite, in conjunction with forsterite, can also be activated by ionizing radiation”. What is meant by “activated”?
- Line 33: Replace “was” with “could” or “was proposed to be”.
- Line 34: Similarly, add “was proposed to” before “played”.
- Figure 1: Font size is too small in places; increase for readability.
- Line 71: Define ATP and AMP.
- Line 72: Clarify whether trimetaphosphate (P_{3m}) refers to its sodium salt. Was this compound chosen as a realistic space analogue, a model compound, or unrelated to expected extraterrestrial chemistry?
- Line 94: Indicate whether radiation was uniform throughout the whole duration of the exposure.
- Line 151: The observed increase in dipeptides in silicate-containing samples could be due to a protective rather than catalytic effect. Stronger evidence for catalysis should be provided. Additionally, address differences between space- and ground-based radiation environments. The differences in radiation levels and types between the space mission and the ground should be addressed at this position. Could these differences be responsible for the observed variation in peptide yield?
- Lines 204–207: The connection drawn to biological systems seems overstated. With limited reactants and processes driven purely by ionizing radiation, the extrapolation to living organisms requires significant caution.
- Line 212: Define EIC in the text, not just in the figure caption.
- Line 286: Reference 34 appears incorrect; please verify.
- Lines 287–289: Clarify how conclusions were drawn from chromatogram comparisons.
- Line 317: The conclusion about radical quenching may not apply to solid-state reactions, where reactants are in direct contact and reactions could be much faster than quenching. Please reconsider this argument.
- Lines 357–358: The claim that these results solve the “phosphorus problem” of life’s origins is unclear. Which pathway do the authors propose, and under what conditions (space or Earth)? The plausibility of the presence of the reactants (in particular, P_{3m}) in this environment should be discussed.

Reviewer #2

(Remarks to the Author)

The paper by Ding et al. describes a plausible chemistry leading to facile synthesis of nucleosides, nucleotides, aminoacylated nucleotides as well as dipeptides triggered by ionizing radiation of extraterrestrial origin. The authors provide evidences that olivine, one of the most ubiquitous minerals present in volcanic rocks exerts a remarkable catalytic effect on these reactions.

While high doses of ionizing radiation are often considered as destructive for the formation of biological building block molecules, the paper demonstrates that prolonged, low-dose radiation might be prosperous for synthetic purposes. I think these are important findings which might deserve to be published in Nature Communications after addressing the following point in a suitably revised version of the manuscript:

Since the detected concentrations are pretty low, they are in the nanomolar range, the authors must present the mass spectra of the untreated reaction mixture in the spectral regions relevant to the detected biomolecular precursors in order to demonstrate that the starting materials do not contain the detected compounds.

Reviewer #3

(Remarks to the Author)

The authors report on radiolysis experiments on the CSS using cosmic radiation to model potential chemistry occurring on asteroidal bodies. The reactions tested were phosphorylation and peptide bond formation. Reactions were also tested using radiolysis facilities on earth to provide evidence for the proposed radiolytic mechanisms. The authors also performed the reactions in the presence of the mineral forsterite, and found it had an important effect on yields. The authors conclude that long-term exposure at low does rates is beneficial compared to the same doses at higher rates.

Overall, this is a fascinating, one-of-a-kind study in radiolytic chemistry, and the fact that radiolysis was carried out on the CSS using cosmic radiation makes this a seminal study. Publication in Nature Communications is recommended after the following comments/criticisms have been addressed.

1. This many seem like a relatively minor comment, but it has important implications. The authors use the phrase “seeds of life” numerous times throughout the manuscript to refer to the canonical amino acid, nucleoside and trimetaphosphate starting materials. Although it's easy to understand why the authors describe these molecules as seeds of life, there is absolutely no evidence that these molecules, especially the ones now used by biology, were actually involved in abiogenesis or were instead the result of Darwinian evolution after life got started. The reviewer suggests to avoid using the phrase “seeds of life” altogether, and use something with stricter scientific meaning.

2. The authors say they use quantification of yields by LCMS, but the exact details of how the yields were determined are not specified either in the main text or the SI. The best way to quantify yields using EICs is to make authentic standards and construct calibration curves with them. The authors need to give more information about how yields were actually determined. For products which have a strong chromophore, like phenylalanine and nucleoside derivatives, detection by UV absorption is another perhaps even more precise alternative.

3. Figure 2. Yields are reported in concentrations with units of nM, but the radiolysis reactions were conducted in the solid state. It is not obvious what concentration is referring to – is it the concentration of the final 5% acetonitrile solution that the samples were dissolved in for analysis? It would be more useful to the reader if the authors converted these concentrations into the amount of moles of product produced. Along similar lines, how much starting material was actually used in each experiment? Section 2 in the SI says consistently “concentration of 30 mmol” or “concentration of 15 mmol”. These are not units of concentration. Do the authors mean mmol/L or do they mean 15 mmol total amounts? The authors really need to use more precise scientific language here.

4. Radiolytic yields. It is typical to report radiolytic yields, i.e, the number of molecules formed per unit of energy absorbed by the sample, usually expressed as number of molecules formed per 100 eV of radiation energy absorbed. Because the way the authors reported the yields, it is not obvious what the yields are in terms of total moles produced, and so the Reviewer is not able to calculate radiolytic yields. The authors should report the radiolytic yields, and provide a discussion that includes comparisons to previously reported values for similar reactions. If the radiolytic yields end up being large compared to similar radiolysis experiments reported previously, then an explanation is needed.

5. The discussion of the mechanism is too unspecific. The fact that addition of TEMPO does not affect yields is interesting, but if no radicals are involved in the mechanism, what exactly is the radiation doing? Are there any other radiolysis studies which demonstrated a similar effect? What makes forsterite special compared to the other minerals tested?

Version 1:

Reviewer comments:

Reviewer #1

(Remarks to the Author)

This version of the manuscript represents a substantial improvement. The authors have satisfactorily addressed all questions raised in my previous review, and I therefore recommend the article for publication.

I have only one minor comment for the authors. In the discussion of the origin of P3M in space (lines 458–462), it is unclear what is specifically meant by the term “space.” Does this refer to extraterrestrial environments associated with planetary bodies (e.g., Enceladus), or to open interplanetary/interstellar space? It should be noted that molecular ejection from Enceladus has already been investigated (Ref. 49 in the manuscript), and no P3M was detected. The authors should therefore first clarify what is meant by “space” in this context. Second, if the authors indeed anticipate the presence of P3M in open space, they should explain how this expectation is reconciled with the previously reported non-detection. For example, do they suggest that the concentration of P3M was below the detection limit of that mission, such that more sensitive future measurements might reveal its presence?

Reviewer #2

(Remarks to the Author)

The authors sufficiently addressed all my comments, therefore I suggest the paper for publication in its present form.

Reviewer #3

(Remarks to the Author)

The authors have responded adequately to all of my criticisms. Publication is now recommended. I suggest that the authors include somewhere in the manuscript or SI the discussion provided in the letter of the measured doses and how it leads to anomalously high G values.

Response point by point

Response for R1

Editorial Office

Nat. Commun.

Dear Editor and reviewers,

Thank you very much for your help and for giving us enough time to complete the manuscript revisions. These reviewers and your comments are constructive in improving our manuscript. After carefully studying the reviewers' comments, we have revised our manuscript based on these comments one by one. All the changes made in the revision were marked in red, and details of this response can be found in the uploaded Word files. Again, we would like to express our great appreciation to you and the reviewers for your comments on our manuscript. If you have any questions, please feel free to contact me again.

As the New Year of 2026 draws near, we extend our warmest wishes to you for a joyous New Year and every success.

Very Sincerely,

Yan Liu, Ph.D

Professor

College of Chemistry and Chemical Engineering, Xiamen University

422 Siming South Road, Xiamen 361005, China.

Tel: +86-592-218-5610. Fax: +86-592-218-6292.

E-mail: stacyliu@xmu.edu.cn

2025-12-26

Reviewer: 1

1.Experiments performed in space.

The exposure experiments conducted onboard the space station require a more detailed description.

Response:

We greatly appreciate your question and suggestion. The sample unit was placed in the Space Radiobiological Exposure Facility (SREF). The radiation dose was mainly monitored by the radiation measurement subsystem. This radiation measurement subsystem consists of four detectors: a radiation linear energy transfer (LET) detector, a neutron spectrum detector, a neutron dose equivalent detector, and a solar ultraviolet irradiation detector. A comprehensive characterization of SREF on the China Space Station has been documented in the published literature¹. All the data we have obtained so far in our experiment are from passive detection. The radiation absorbed dose obtained in this experiment was measured by a radiation thermoluminescence detector and mainly consisted of radiation with LET values less than 10 keV/ μm , including protons, neutrons and some heavy ion radiation and so on.

To investigate space ionizing radiation effects, we specially designed two types of experimental unit sample boxes. One is made of unopened window polyimide material, which can accept ionizing radiation while shielding UV. Another one is made of stainless-steel material and features a quartz glass window. It can be used to investigate the coupling effect between space ionizing and UV radiation. In our present work, space ionizing radiation effects are the focus of our attention. By presetting various passive radiation detection chips in the sample box of each experimental unit, we obtained the average total radiation dose sensed by each layer of the experimental unit. The above-mentioned exposure experimental details, including the radiation environment, temperature, pressure, atmosphere, etc., have been supplemented into the revised Supplementary Information and marked in red.

References:

1. Zhang B, *et al.* The Space Radiobiological Exposure Facility on the China Space Station. *Astrobiology* **25**, 32-41 (2025).

2. What types of radiation reached the samples, and which are expected to play the dominant role in the studied processes?

Response:

Thank you for your good question. As mentioned above, space radiation experiment samples in the present work placed in the polyimide experiment unit of the Space Radiobiological Exposure Facility were exposed to various types of space ionizing radiation, such as galactic cosmic rays, protons, electrons, occasional high-energy solar particles, secondary neutrons, and X-rays. Due to the shielding effect of polyimide material, these experimental samples have not been exposed to UV radiation.

3. Were all samples exposed to the same dose of radiation, or could their positioning have caused shielding effects?

Response:

Thank you very much for your good question. The radiation doses sensed by different sites on the same layer of samples are not significantly different, with only slight variations. There are some obvious deviations between different sample layers. Variations in radiation dose were observed across different measurement positions, the details were shown in Supplementary Table 1. All the values presented in this work are therefore based on the calculated average dose to ensure a consistent frame of reference. The parallel samples and the control group involved in our experiment were placed on the same layer to facilitate comparison and discussion.

Supplementary Figure 1 The layout of each layer of the polyimide sample box (right) and the passive detection chips (Left, chips marked with the numbers)

Supplementary Table 1. The irradiation measurement detected by passive detection chips

Radiation Dose (mGy)		Detection Chip	1	2	3	4	5	6
Layer of Sample Box								
TZ-6	B3-A [#]		78.9	73.2	73.4	71.2	73.2	71.5
	B3-B [#]		82.8	78.3	74.7	74.6	76.3	77.1
	B3-C [#]		74.1	81.4	78.9	75.8	84.8	81.3
	B3-D [#]		95.6	86.7	83.7	85.9	88.8	81.2
	B3-E [#]		103.9	105.9	98.8	105.9	98.8	98.0
	C3-A [#]		33.3	31.6	36.0	35.6	33.7	31.4
	C3-B [#]		31.7	27.7	32.6	37.6	32.9	32.5
	C3-C [#]		35.4	28.5	35.4	37.5	35.3	34.0
	C3-D [#]		40.9	33.1	41.6	39.4	40.1	41.1
	C3-E [#]		44.5	37.8	39.5	45.4	48.7	47.9

Notes: Numbers from 1 to 6 are the serial numbers of the passive detection chips in each layer (**Supplementary Figure 1d** (left)). [#]: The serial numbers of each layer of the sample box.

4. What atmosphere was present above the samples, why was it chosen, and what natural conditions does it aim to replicate?

Response:

Thank you for your insightful question. We conducted exposure experiments in two distinct atmospheric environments, specifically air and argon atmospheres, which served as mutual references. Active oxygen species, including oxygen molecules, have been reported in interstellar space^{2, 3}. Oxygen molecules can generate various reactive oxygen species under the irradiation of space radiation. To explore the effects of active oxygen species on

our reaction systems, we designed and conducted our space expose experiments in an air environment. The argon experiment groups were used as references to explain the results of the dehydration condensation reaction without active oxygen species components.

We discovered that when argon was substituted for air, in addition to dipeptide and nucleoside monophosphate, more complex aminoacyl adenylate analogues could all be detected effectively (Supplementary Figure 128) in the same reaction system. That indicates that regardless of whether it is in an air (active oxygen environment) or an argon atmosphere (oxygen free environment), the formation of peptides and phosphoester bonds can also be effectively achieved. This finding suggests a broader and more resilient environmental context for the chemical origins of life.

Supplementary Figure 1. The extracted ion chromatogram (EIC) of reaction products (5'-AMP, 3'-AMP, 5'-N-Phe-AMP, and Phe-Phe) formed from the reaction system containing Phe, nucleosides (A and U), and P₃m with forsterite under the CSS radiation with a total dose of 53.50 mGy in A2 Unit (argon atmosphere) of the TZ 7 launch mission (extravehicular exposure for 174 days) and without radiation on the ground. The results also demonstrate that no related products were detected in the absence of radiation.

The aforementioned discussion has been incorporated into the revised manuscript (in the section regarding the mechanism investigation).

References

2. Bieler A, *et al.* Abundant molecular oxygen in the coma of comet 67P/Churyumov–Gerasimenko. *Nature* **526**, 678-681 (2015).

3. Yen AS, Kim SS, Hecht MH, Frant MS, Murray B. Evidence That the Reactivity of the Martian Soil Is Due to Superoxide Ions. *Science* **289**, 1909-1912 (2000).

5. Literature review.

The introduction should more thoroughly review prior research on peptide formation, particularly in the solid state (<https://doi.org/10.1038/ncomms9385>, <https://doi.org/10.1039/C9RA00179D>) and in astrophysical environments (e.g., <https://doi.org/10.21203/rs.3.rs-6939044/v1>, <https://doi.org/10.1126/sciadv.adj7179>, <https://doi.org/10.1038/s41550-021-01577-9>, <https://doi.org/10.1088/0004-637x/765/2/111>).

Response:

Thank you very much for the constructive suggestion. The relevant literature highlighted by the reviewers is instrumental in guiding our understanding of peptide formation in outer space and serves as key references for theorizing its underlying mechanisms. Dehydration-hydration cycles enable the high-yield formation of long oligopeptides from unactivated amino acids under simple, programmable conditions, offering a plausible mechanism for prebiotic peptide synthesis. (Rodriguez-Garcia, M., Surman, A., Cooper, G. et al. Formation of oligopeptides in high yield under simple programmable conditions. *Nat Commun* **6**, 8385 (2015). <https://doi.org/10.1038/ncomms9385>). Through experimental simulation, this research (<https://doi.org/10.21203/rs.3.rs-6939044/v1>, Preprint) demonstrates that Gly-gly dipeptide can solid-state polymerized in interstellar ice analogues exposed to ionizing radiation.

Besides, previous studies have reported that the solid-liquid interface in an electrochemical system serves as a platform for amino acid polymerization, facilitating the formation of poly-glycine films featuring peptide bonds⁴. The related experimental data verify that peptide synthesis can be achieved under simulated astrophysical conditions, even without water. In addition, other reported key pathways of solid-state peptide polymerization also include the barrierless condensation of atomic carbon^{5, 6} and radiation-induced solid-state polymerization⁷.

The above-mentioned literature has been supplemented into our revised main manuscript, which is marked in red.

References

4. Ali MF, Abdel-Aal FA. In situ polymerization and FT-IR characterization of poly-glycine on pencil graphite electrode for sensitive determination of anti-emetic drug, granisetron in injections and human plasma. *RSC Adv.* **9**, 4325-4335 (2019).
5. Krasnokutski SA, Chuang KJ, Jäger C, Ueberschaar N, Henning T. A pathway to peptides in space through the condensation of atomic carbon. *Nat. Astron.* **6**, 381-386 (2022).
6. Krasnokutski SA, Jäger C, Henning T, Geffroy C, Remaury QB, Poinot P. Formation of extraterrestrial peptides and their derivatives. *Sci. Adv.* **10**, eadj7179 (2024).
7. Kaiser RI, Stockton AM, Kim YS, Jensen EC, Mathies RA. On the formation of dipeptides in interstellar model ices. *Astrophys. J.* **765**, 111 (2013).

6. Stability of biomolecules under radiation

The potentially most compelling aspect of these experiments — the stability of biomolecules under ionizing radiation - is not addressed. Did the authors assess the degradation of dipeptides or other molecules? This information would be valuable in itself and especially relevant for interpreting differences in molecular abundance across samples.

Response:

Thank you very much for your good question. The radiation absorbed dose obtained in this experiment was measured by a radiation thermoluminescence detector and mainly consisted of radiation with LET values less than 10 keV/μm, including protons, neutrons and some heavy ion radiation.

Through our space expose experiments, we evaluated the photochemical stability of different chiral configuration Phe-Phe dipeptide (*L*-Phe-*L*-Phe, *L*-Phe-*D*-Phe and *D*-Phe-*D*-Phe) by HPLC analysis. The related experimental results revealed that Phe-Phe dipeptide

exhibited remarkable stability in the outboard radiation environment of the CSS, showing no detectable degradation. The photochemical stability of different chiral configuration Phe-Phe dipeptide doesn't exhibit the significant difference. The above-mentioned experimental results have been exhibited in the revised Supplementary Information as **Supplementary Figure 169**.

Supplementary Figure 169. The photochemical stability of different chiral configuration Phe-Phe dipeptide by HPLC quantitative analysis. These dipeptides were exposed to the outboard radiation environment of the CSS with a total dose of 37.75 mGy without minerals in C3 Unit.

The aforementioned discussion has been incorporated into the Discussion Section of the revised manuscript.

Minor Issues

- **Line 21: Define AMP.**

Response:

Thanks a lot for your kind remind. The definition of AMP has been supplemented into the revised manuscript, which was marked in red (Line 21 of the revised manuscript).

- **Lines 21–22: Clarify the phrase “Hydroxyapatite, in conjunction with forsterite, can also be activated by ionizing radiation”. What is meant by “activated”?**

Response:

Thanks a lot for your kind remind. Here, we want to express that “Ionizing radiation combined with the aid of forsterite can promote hydroxyapatite to activate amino acids to form dipeptides.” We have revised the expression in our revised manuscript as “Under ionizing radiation, forsterite can promote the utilization of hydroxyapatite in prebiotic chemistry. In this context, hydroxyapatite can act as an accessible phosphorus source to activate amino acids for peptide formation.” These revisions were marked in red in the revised manuscript).

• **Line 33: Replace “was” with “could” or “was proposed to be”.**

Response:

Thanks a lot. According to your suggestions, these inappropriate parts have been revised and marked in red in the revised manuscript.

• **Line 34: Similarly, add “was proposed to” before “played”.**

Response:

Yes. According to your suggestions, these inappropriate parts have been revised and marked in red in the revised manuscript.

• **Figure 1: Font size is too small in places; increase for readability.**

Response:

Thanks a lot. According to your suggestions, font size of Figure 1 has been revised in the revised manuscript.

• **Line 71: Define ATP and AMP.**

Response:

Thanks a lot for your kind remind. The definition of AMP and ATP have been supplemented into the revised manuscript, which were marked in red (Line 100~101 of the revised manuscript).

• Line 72: Clarify whether trimetaphosphate (P_{3m}) refers to its sodium salt. Was this compound chosen as a realistic space analogue, a model compound, or unrelated to expected extraterrestrial chemistry?

Response:

Thank you very much for the excellent question. Trimetaphosphate (P_{3m}) used and discussed in this paper is sodium trimetaphosphate (P_{3m}). We have supplemented its cationic form into the main manuscript and Supplementary Information.

Sodium trimetaphosphate (P_{3m}) is widely recognized as a soluble phosphorus source that can be effectively utilize in prebiotic environments. Previous literature indicates that sodium trimetaphosphate can be synthesized either by condensing monophosphates in an aqueous solution⁸ or through a reaction with diamidophosphate (DAP)⁹.

Recent research has reported that heat flows can solubilize apatite, subsequently producing P_{3m} and then enhancing phosphate availability for prebiotic chemistry¹⁰. In space environments, especially in the Enceladus's ocean, huge phosphate sources¹¹, suitable temperature¹² and similar heat flow system¹³ have been found and confirmed. Therefore, we speculate that P_{3m} may also exist in space, despite the absence of reports to date. Combined with our experimental experience in the study of the P_{3m} system, we selected P_{3m} as the soluble and effective phosphorus reagent in our chemistry reaction model.

The discussion mentioned above has been incorporated into the revised main manuscript (Lines 93~94, and Lin 460-465).

Reference:

8. Chu XY, Zhang HY. Prebiotic Synthesis of ATP: A Terrestrial Volcanism-Dependent Pathway. *Life (Basel)* **13**, 731 (2023).

9. Osumah A, Krishnamurthy R. Diamidophosphate (DAP): A Plausible Prebiotic Phosphorylating Reagent with a Chem to BioChem Potential? *Chembiochem* **22**, 3001-3009 (2021).
10. Matreux T, *et al.* Heat flows solubilize apatite to boost phosphate availability for prebiotic chemistry. *Nat. Commun.* **16**, 1809 (2025).
11. Postberg F, *et al.* Detection of phosphates originating from Enceladus's ocean. *Nature* **618**, 489-493 (2023).
12. Hsu H-W, *et al.* Ongoing hydrothermal activities within Enceladus. *Nature* **519**, 207-210 (2015).
13. Waite JH, *et al.* Cassini finds molecular hydrogen in the Enceladus plume: Evidence for hydrothermal processes. *Science* **356**, 155-159 (2017).

• **Line 94: Indicate whether radiation was uniform throughout the whole duration of the exposure.**

Response:

Thank you very much for the good question. Space ionizing radiation is not uniform throughout the whole duration of the exposure. In our experiment unit, the radiation doses sensed by different sites on the same layer of samples are not significantly different, with only slight variations. There are some obvious deviations between different sample layers. Variations in radiation dose were observed across different measurement positions, the details were shown in Supplementary Table 1. The three groups of parallel samples and the control group on the CSS involved in our experiment were placed on the same layer to facilitate comparison and discussion. All the the values presented in this work are the calculated average dose to ensure a consistent frame of reference and guide the setting of radiation doses for ground control groups.

• **Line 151: The observed increase in dipeptides in silicate-containing samples could be due to a protective rather than catalytic effect. Stronger evidence for catalysis should be provided. Additionally, address differences between space- and ground-**

based radiation environments. The differences in radiation levels and types between the space mission and the ground should be addressed at this position. Could these differences be responsible for the observed variation in peptide yield?

Response:

Thank you very much for your good question. Onboard the CSS, the radiation stems from the coupling effect of multiple types of space ionizing radiation. Based on the radiation dose calculated by the preset passive detection chips, we established control groups using X-ray, a single radiation source.

First, as shown in Fig. 2e, the yield of AMP in the experimental system exposed to low-dose space ionizing radiation without forsterite for 174 days was higher compared to that exposed for 97 days. However, the yield declined after a 294-day space exposure. This indicates that as the radiation time extends, the amount of AMP generated in the initial stage initially increases. As the radiation time is further extended, the yield of AMP decreases, and photodegradation of AMP takes place. This experimental phenomenon indicates that low-dose space ionizing radiation can promote the dehydration condensation reaction, such as the phosphorylation of nucleosides. However, excessive radiation exposure also induces the photodegradation reaction (Fig. 2c and d).

When forsterite was introduced into the space exposure experiment system, in the presence of forsterite minerals, the production AMP was 3.8 times higher than that in the absence of forsterite. This indicates that the presence of forsterite has a promoting effect on the formation of products. Since our reaction system involves a solid-phase reaction, the concentration and enrichment effect of forsterite via electrostatic interaction will not play a significant role. This promoting effect maybe come from the radiation resistance of forsterite or catalytic action.

To further explain the role of forsterite, we carried out a series of experiments on the ground. Upon the addition of the metal chelating agent ethylenediaminetetraacetic acid (EDTA) to the reaction system, the yield of the dipeptide decreased by approximately twelve-fold, which confirms the significant involvement of magnesium ions (Fig. 5c).

This conclusion was further corroborated by comparing different mineral substrates: the use of magnesium oxide (MgO) resulted in the highest dipeptide yield (7801.50 pmol), surpassing that obtained with silicon dioxide (SiO₂, the dipeptide yield 454.02 pmol) or forsterite (the dipeptide yield 6217.20 pmol, Fig. 5d). It further indicates that the main catalytic component in forsterite is Mg²⁺.

In addition, the pH value of the reaction system containing forsterite is alkaline (pH = 8.06, Fig. 5e), which is conducive to the formation of peptide bonds and phosphoester bonds.

Fig. 5 Exploration of the mechanism of Photochemical Peptide Bond Formation. a. Trapping of reactive radical species with TEMPO, leading to the formation of stable alkoxyamine adducts for mechanistic elucidation. b. The peak area analysis results of the extracted ion chromatogram (EIC) of the product Phe-Phe at *m/z* 313.1547 from the forsterite-solid phase with/without TEMPO in the Phe systems containing P₃m and A/U reaction system with 63.63 mGy (17 s) X-ray radiation. c. Quantification analysis of Phe-Phe production with forsterite with/without EDTA in the Phe systems containing P₃m

reaction system with 63.63 mGy (17 s) X-ray radiation. d. Quantification analysis of Phe-Phe production with different mineral in the Phe systems containing P₃m reaction system with 63.63 mGy (17 s) X-ray radiation. e. The Phe, P₃m and C/G solutions at different pH values of various minerals. ** $p < 0.01$, *** $p < 0.0001$, ns: not significant. Bars represent mean values, and error bars indicate mean \pm SD (All experiments were performed with three experimental replicates. Statistical outliers were excluded from the final analysis. $n = 2\sim 3$ independent experiments).

In conclusion, apart from the radiation resistance effect, which is a common function of minerals, we propose that the characteristic functions of forsterite in our reaction system primarily encompass the following three aspects: 1) to offer an alkaline environment, which facilitates the formation of peptide bonds and phosphoester bonds; 2) adsorption induced by electrostatic interaction; 3) the catalytic effect centered around magnesium ions.

The discussion mentioned above has been incorporated into the revised main manuscript.

• Lines 204–207: The connection drawn to biological systems seems overstated. With limited reactants and processes driven purely by ionizing radiation, the extrapolation to living organisms requires significant caution.

Response:

Thank you very much for your helpful suggestion. According to your suggestion, we have revised the relevant expressions in the revised manuscript through deleting the related statement, such as “The dominant generation of AMP in the given competitive reaction system may imply nature's selection biases of adenosine to meet the generation of energy currency and cofactors, ATP or NADP, during the evolution process of life.”

• Line 212: Define EIC in the text, not just in the figure caption.

Response:

We are grateful to you for pointing out this oversight. We have supplemented the definition of EIC into the revised manuscript at the Line 142.

- **Line 286: Reference 34 appears incorrect; please verify.**

Response:

We are grateful to you for reminding us of this oversight. We have amended the correct reference in the revised manuscript.

- **Lines 287–289: Clarify how conclusions were drawn from chromatogram comparisons.**

Response:

Thank you very much for your good question. Based on our previous work, which demonstrated that Phe, P_{3m}, and adenine (A) can form the *N*-nucleotide amidate (*N*-Phe-AMP) in alkaline aqueous solution—with the 2'-*N*-Phe-AMP isomer characterized as the primary product—we applied the same analytical methodology to the reaction system containing minerals. At the same time, the resulting MS extraction ion chromatograms were analyzed and compared with each other. Since we know the retention characteristics of 2'-*N*-Phe-AMP, the identification of the observed species in solid-phase reaction system was carried out based on a comparison of their chromatographic retention times.

- **Line 317: The conclusion about radical quenching may not apply to solid-state reactions, where reactants are in direct contact and reactions could be much faster than quenching. Please reconsider this argument.**

Response:

Thank you very much for your good question. To ensure the uniformity and availability of the reactants' contact, 3 equivalents of the radical quenching agent 2, 2, 6, 6-tetramethylpiperidinoxy (Tempo) solution were used and dropped into the ground control Phe reaction system. The resulting reaction systems were freeze-dried and then exposed to X-rays. The radical trapping process by Tempo is depicted in the Fig. 5a. Three parallel tests were carried out. The experimental results indicated that the yield of the key dipeptide product did not decrease (Fig. 5b and Supplementary Figs. 141). At the same time, inspired by the literature, we analyzed and detected the free radical products captured by

Tempo relating to the amino acids by HRMS (Supplementary Figs. 142-145). It indicates that the amide and phosphoester bonds formed in the reaction system can undergo radical reactions, yet this is not the sole pathway. The related reactions predominantly proceed via an ionic reaction pathway.

• **Lines 357–358: The claim that these results solve the “phosphorus problem” of life’s origins is unclear. Which pathway do the authors propose, and under what conditions (space or Earth)? The plausibility of the presence of the reactants (in particular, P_{3m}) in this environment should be discussed.**

Response:

Thank you very much for your constructive questions. According to your question, we have supplemented the following discussion in the revised manuscript.

“Apatite is the primary form of phosphorus resources, not only on Earth but also present across the space, such as Mars. However, apatite is insoluble in water, a characteristic that limits its application in prebiotic chemistry. In this work, after replacing P_{3m} with hydroxyapatite in our reaction system on the ground, we observed that hydroxyapatite can be utilized to promote peptide formation under the combined effects of ionizing radiation and forsterite (Supplementary Figs. 147-159). These experimental results offer a new approach to addressing the "phosphorus limitation" in the origin of life.”

“Recent research has reported that heat flows can solubilize apatite, subsequently producing P_{3m} and then enhancing phosphate availability for prebiotic chemistry. In space environments, especially in the Enceladus, huge phosphate sources, suitable temperature and similar heat flow system have been found and confirmed. Therefore, we speculate that P_{3m} may also exist in space, despite the absence of reports to date.”

Answers to the above related questions has also been explained in the response to the question about the **Line 72**.

Reviewer 2:

Since the detected concentrations are pretty low, they are in the nanomolar range, the authors must present the mass spectra of the untreated reaction mixture in the spectral regions relevant to the detected biomolecular precursors in order to demonstrate that the starting materials do not contain the detected compounds.

Response:

Thanks for this constructive advice. For comparison, non-irradiated initial raw materials and control groups were also analyzed by MS (Supplementary Figs. 42, 128). For example, we analyzed the initial raw materials of the reaction system and did not observe the existence of dipeptides (Black line in Supplementary Fig. 42). It further indicated that long-term low-dose ionizing radiation can induce the formation of peptides with the assistance of forsterite.

Figure S42. The extracted ion chromatogram (EIC) of reactant raw materials (Phe and Ala) and reaction product (e.g., Phe-Arg) dipeptide formed in the mixed amino acid (Ala, Phe, and Arg) reaction systems setting up in TZ 6 Unit B3 with the assistance of forsterite and a total dose of 101.08 mGy onboard the CSS, with the assistance of forsterite and a total dose of 10 Gy on the ground, without forsterite and a total dose of 10 Gy on the ground and with the assistance of forsterite but without radiation on the ground. The relevant products were detectable in all ground samples, irrespective of the presence of radiation or forsterite.

Supplementary Figure 128. The extracted ion chromatogram (EIC) of reaction product (5'-AMP, 3'-AMP, 5'-N-Phe-AMP and Phe-Phe) product formed from the reaction system containing Phe, nucleosides (A and U) and P₃m with forsterite under the CSS radiation with a total dose of 53.50 mGy in A2 Unit of the TZ 7 launch mission (extravehicular exposure for 174 days) and without radiation on the ground. The results demonstrate that no related products were detected in the absence of radiation.

The above-mentioned explanation has been supplemented in the revised manuscript.

Reviewer 3:

1. This may seem like a relatively minor comment, but it has important implications. The authors use the phrase “seeds of life” numerous times throughout the manuscript to refer to the canonical amino acid, nucleoside and trimetaphosphate starting materials. Although it's easy to understand why the authors describe these molecules as seeds of life, there is absolutely no evidence that these molecules, especially the ones now used by biology, were actually involved in abiogenesis or were instead the result of Darwinian evolution after life got started. The reviewer suggests to avoid using the phrase “seeds of life” altogether, and use something with stricter scientific meaning.

Response:

Thank you very much for your kind reminder and constructive suggestion. According to your suggestion, “seeds of life” has been replaced with “Prebiotic organic molecules” in the revised manuscript.

2. The authors say they use quantification of yields by LCMS, but the exact details of how the yields were determined are not specified either in the main text or the SI. The best way to quantify yields using EICs is to make authentic standards and construct calibration curves with them. The authors need to give more information about how yields were actually determined. For products which have a strong chromophore, like phenylalanine and nucleoside derivatives, detection by UV absorption is another perhaps even more precise alternative.

Response:

Thank you very much for this helpful advice. Throughout the entire research work, we first utilized the combined technology of high-resolution mass spectrometry (ESI-Orbitrap-HRMS) and ultra-high pressure chromatography (UPLC) to achieve qualitative analysis and identification of potential target products.

Secondly, based on the excellent stability and reproducibility of ESI-Triple quadrupole (TQ) mass spectrometry, we conducted the quantitative analysis for the characteristic target products by TQ-MS (AB SCIEX QTrapTM 5500 LC-MS). We dissolved dipeptide standards (namely Ala-Ala, Ala-Phe, Phe-Phe, Phe-Ala and Phe-Arg) and NMP standards (AMP, UMP, CMP and GMP) in pure water to prepare mother liquors. Subsequently, these mother liquors were diluted to five different concentrations and analyzed using AB SCIEX QTrapTM 5500 LC-MS. Calibration curves were then constructed based on the analysis results. The related quantitative parameters and standard curves have been added in the revised Supplementary Information (Supplementary Table 4 and Supplementary Figs 160-168).

Supplementary Table 4. Mass parameters for the quantitative analysis of dipeptide and NMP by AB

SCIEX QTrapTM 5500.

Analytes	Q1 Mass (Da)	Q3 Mass (Da)	Dwell time (msec)	DP (V)	CE (V)
Phe-Phe	313.1	120.1*	50	48	28
		166.1	50	48	19.5
Ala-Ala	161.0	90.0*	50	10	13

		115.0	50	10	12
Phe-Ala	237.1	120.1*	50	20	20
		130.1	50	20	50
Phe-Arg	322.1	175.1*	50	70	29
		120.1	50	70	37
Ala-Phe	237.1	166.1*	50	25	17
		120.1	50	25	28
AMP	348.1	136.0*	20	70	24
UMP	323.0	79.0*	20	-80	-52
		210.9	20	-80	-23
CMP	322.0	79.0*	20	-110	-25
		211.0	20	-110	-23
GMP	362.0	79.0*	20	-100	-27
		211.0	20	-100	-25

Notes: * means the fragment ions used for quantitative analysis and the rest for qualitative analysis. DP, declustering potential; CE, collision energy.

Supplementary Figure 160. Calibration curve of the Ala-Ala.

Supplementary Figure 161. Calibration curve of the Ala-Phe.

Supplementary Figure 162. Calibration curve of the Phe-Phe.

Supplementary Figure 163. Calibration curve of the Phe-Ala.

Supplementary Figure 164. Calibration curve of the Phe-Arg.

Supplementary Figure 165. Calibration curve of the AMP.

Supplementary Figure 166. Calibration curve of the UMP.

Supplementary Figure 167. Calibration curve of the CMP.

Supplementary Figure 168. Calibration curve of the GMP.

For the quantification of each sample batch, a new standard curve shall be established, as illustrated in the above example.

3. Figure 2. Yields are reported in concentrations with units of nM, but the radiolysis reactions were conducted in the solid state. It is not obvious what concentration is referring to – is it the concentration of the final 5% acetonitrile solution that the samples were dissolved in for analysis? It would be more useful to the reader if the

authors converted these concentrations into the amount of moles of product produced. Along similar lines, how much starting material was actually used in each experiment? Section 2 in the SI says consistently “concentration of 30 mmol” or “concentration of 15 mmol”. These are not units of concentration. Do the authors mean mmol/L or do they mean 15 mmol total amounts? The authors really need to use more precise scientific language here.

Response:

We are grateful to you for pointing out this oversight. According to your suggestion, the corresponding revisions have been made in the revised manuscript. As shown in the revised Fig 2, the quantity of the product is now represented as the number of moles of the product produced instead of its concentration.

Fig. 2 Quantitative analysis of the products using an AB SCIEX QTrap™ 5500 LC-MS instrument. **a**, Amounts of product dipeptides obtained from the mixed amino acid system containing Phe, Ala, and Arg onboard the CSS. **b**, Relative contents of the product dipeptides in **Fig. 2a**. The levels of Ala-Ala and Phe-Ala in the control group (without forsterite) were below the quantification limit. **c**, Quantification analysis of AMP production in the Phe, Ala and Arg systems containing P₃m and A/U but without forsterite onboard the CSS (Unit C3 and B3). A/U stands for this reaction system contains A and U. C/G stands for this reaction system contains C and G. **d**, Quantification analysis of NMP (AMP, UMP, CMP and GMP) produced in the Arg systems containing P₃m and A/U but without forsterite

onboard the CSS (Unit C3 and B3). **e**, Quantification analysis of AMP production in the Phe systems containing P₃m and A/U with forsterite onboard the CSS (Unit A2) and without forsterite onboard the CSS (Unit C3, B3 and A2) and the control groups exposed to varying radiation doses. $^{\&}p < 0.0001$ vs. the TZ 7 sample with forsterite, $^{\#}p < 0.0001$ vs. the TZ 6 of Unit C3 sample. **f**, Extracted ion chromatogram (EIC) of the product AMP (m/z 348.0704) in the reaction system of **Fig. 2e** set up in Unit C3 receiving a total radiation dose of 37.75 mGy onboard the CSS. 5'-AMP-STD and 3'-AMP-STD were used as standards. The peak at 2.96 min corresponds to 5'-AMP with approximately 96% regioselectivity. **g**, Quantification analysis of Phe-Phe produced in the different Phe systems onboard the CSS (Unit A2) and the ground control under 20 Gy of radiation for 240 s. Blank stands for without forsterite. $^*p < 0.0001$ vs. the TZ 7 sample with forsterite, $^{\wedge}p < 0.0001$ vs. the ground sample with forsterite. Statistical analysis was conducted using GraphPad Prism version 8.0. Bars represent mean values, and error bars indicate mean \pm SD (All experiments were performed with three experimental replicates. Statistical outliers were excluded from the final analysis. $n = 2-3$ independent experiments).

4. Radiolytic yields. It is typical to report radiolytic yields, i.e, the number of molecules formed per unit of energy absorbed by the sample, usually expressed as number of molecules formed per 100 eV of radiation energy absorbed. Because the way the authors reported the yields, it is not obvious what the yields are in terms of total moles produced, and so the Reviewer is not able to calculate radiolytic yields. The authors should report the radiolytic yields, and provide a discussion that includes comparisons to previously reported values for similar reactions. If the radiolytic yields end up being large compared to similar radiolysis experiments reported previously, then an explanation is needed.

Response:

Thank you for your enlightening advice. The G -value quantifies the radiation chemical yield, representing the number of molecular entities formed or destroyed per 100 eV of energy deposited in an irradiated system¹. G -value can be calculated according to the equation (1).

$$G = N/E_{\text{abs}}(\text{eV}) \times 100 \quad (1)$$

Here, $N=n \times N_A$; $E_{\text{abs}}=D \times m$. D (Gray, Gy), 1 Gy= 1 J/Kg. 1 Gy= 6.242×10^{18} eV/Kg

N : The amount of molecules; n : total molar amount of the target molecule produced (or consumed) in the irradiated sample; N_A : Avogadro constant, $6.02214076 \times 10^{23}$ mol⁻¹; m : Total mass of the irradiated sample; D : Average absorbed dose of the sample; $E_{\text{abs}}(\text{eV})$: Total radiant energy absorbed.

Taking the Phe-Arg produced under forsterite-present conditions in sample TZ 6 (Table 1) as an example, irradiation with a total dose of 37.75 mGy resulted in a yield of 29.39 pmol. The total mass of the radiated samples was approximately 50 mg, with a corresponding G -value of 150 molecules per 100 eV.

$$\begin{aligned} G &= [N/E_{\text{abs}}(\text{eV})] \times 100 \\ &= [29.39 \times 10^{-12} \times 6.02 \times 10^{23} / 37.75 \times 10^{-3} \times 6.242 \times 10^{18} \text{ eV/Kg} \times 50 \times 10^{-3} \times 10^{-3} \text{ Kg}] \times 100 \\ &= [29.39 \times 6.02 \times 10^{11} / 37.75 \times 6.242 \times 50 \times 10^9] \times 100 \\ &= 150 \end{aligned}$$

The relatively high G -value calculation may be attributed to the reported dose of 37.75 mGy in this reaction system on the CSS (Supplementary Figure 1c, lithium 6 or 7 passive detection chips) mainly corresponds to low-LET radiation (<10 keV/ μm), comprising protons, electrons, thermal neutrons, and X/ γ rays without accounting for high-LET heavy ions. Radiolytic yields with heavy ions are complex, and they depend on both LET and nuclear charge². The space radiation environment is extremely complex. Since the direction of the LET in space is random and there is no definite incident direction, current detectors cannot measure in all directions. Instead, they can only provide measurements for certain directions, and these measurements are incomplete.

Given that the G -value here does not represent a completely accurate situation, we did not provide the G -value for each product.

Supplementary Figure 1. The schematic diagram of the Space Radiobiological Exposure Facility. a. The overall scheme of the experiment device; b. The single sample container (Polyimide material); c. The layout of passive detection chips in the sample unit. d. The layout of each layer of the polyimide sample box (right) and the passive detection chips (left).

References

1. Baird JK, Miller GP, Li N. The G value in plasma and radiation chemistry. *J. Appl. Phys.* **68**, 3661-3668 (1990).
2. LaVerne JA, Chang Z, Araos MS, Heavy ion radiolysis of organic materials, *Radiat. Phys. Chem.* **60**, 253-257 (2002).

5. The discussion of the mechanism is too unspecific. The fact that addition of TEMPO does not affect yields is interesting, but if no radicals are involved in the mechanism, what exactly is the radiation doing? Are there any other radiolysis studies which demonstrated a similar effect? What makes forsterite special compared to the other minerals tested?

Response:

Thank you very much for your constructive question. To further investigate the underlying mechanism and the role of forsterite, we have carried out a series supplementary experiments, containing the radical scavenger (TEMPO) addition experiment, metal ion chelating reagent (EDTA) addition experiment, and pH determination experiment of the reaction system. Relevant experimental results and discussions have been supplemented in the **“Molecular mechanism of ionizing radiation-induced condensation and phosphorylation reaction”** section of the revised manuscript.

To ensure the uniformity and availability of the reactants' contact, 3 equivalents of the radical quenching agent 2, 2, 6, 6-tetramethylpiperidinoxy (TEMPO) solution were used and dropped into the ground control Phe reaction system. The resulting reaction systems were freeze-dried and then exposed to X-rays. The radical trapping process by TEMPO is depicted in the Fig. 5a. Three parallel tests were carried out. The experimental results indicated that the yield of the key dipeptide product did not decrease (Fig. 5b and Supplementary Figs. 141). At the same time, inspired by the literature, we analyzed and detected the free radical products captured by TEMPO relating to the amino acids by HRMS (Supplementary Figs. 142-145). It indicates that the amide and phosphoester bonds formed in the reaction system can undergo radical reactions, yet this is not the sole pathway. The related reactions predominantly proceed via an ionic reaction pathway.

To further explain the role of forsterite, we carried out a series of experiments on the ground. Upon the addition of the metal chelating agent ethylenediaminetetraacetic acid (EDTA) to the reaction system, the yield of the dipeptide decreased by approximately twelve-fold, which confirms the significant involvement of magnesium ions (Fig. 5c).

This conclusion was further corroborated by comparing different mineral substrates: the use of magnesium oxide (MgO) resulted in the highest dipeptide yield (7801.50 pmol), surpassing that obtained with silicon dioxide (SiO₂, the dipeptide yield of 454.02 pmol) or forsterite (the dipeptide yield of 6217.20 pmol, Fig. 5d). It further indicates that the main catalytic component in forsterite is Mg²⁺.

In addition, the pH value of the reaction system containing forsterite is alkaline (pH = 8.06, Fig. 5e), which is conducive to the formation of peptide bonds and phosphoester bonds.

Fig. 5. Exploration of the mechanism of Photochemical Peptide Bond Formation. a. Trapping of reactive radical species with Tempo, leading to the formation of stable alkoxyamine adducts for mechanistic elucidation. b. The peak area analysis results of the extracted ion chromatogram (EIC) of the product Phe-Phe at m/z 313.1547 from the forsterite-solid phase with/without Tempo in the Phe systems containing P₃m and A/U reaction system with 63.63 mGy (17 s) X-ray radiation. c. Quantification analysis of Phe-Phe production with forsterite with/without EDTA in the Phe systems containing P₃m reaction system with 63.63 mGy (17 s) X-ray radiation. d. Quantification analysis of Phe-Phe production with different minerals in the Phe systems containing P₃m reaction system with 63.63 mGy (17 s) X-ray radiation. e. The Phe, P₃m and C/G solutions at different pH values of various minerals. **: $p < 0.01$, ****: $p < 0.0001$, ns: not significant. Bars represent mean values, and error bars indicate mean \pm SD (All experiments were performed with three experimental replicates. Statistical outliers were excluded from the final analysis. $n = 2\sim 3$ independent experiments).

In conclusion, inspired by the comments of the two reviewers and based on the additional experimental results, we propose that the amide and phosphoester bonds formed in the reaction system can undergo radical reactions; however, this is not the only pathway. The related reactions mainly proceed through an ionic reaction pathway.

Apart from the radiation resistance effect, which is a common function of minerals, we propose that the characteristic functions of forsterite in our reaction system primarily encompass the following three aspects: 1) to offer an alkaline environment, which facilitates the formation of peptide bonds and phosphoester bonds; 2) adsorption induced by electrostatic interaction; 3) the catalytic effect centered around magnesium ions.

Response for R2

Reviewer 1

This version of the manuscript represents a substantial improvement. The authors have satisfactorily addressed all questions raised in my previous review, and I therefore recommend the article for publication.

I have only one minor comment for the authors. In the discussion of the origin of P₃M in space (lines 458–462), it is unclear what is specifically meant by the term “space.” Does this refer to extraterrestrial environments associated with planetary bodies (e.g., Enceladus), or to open interplanetary/interstellar space? It should be noted that molecular ejection from Enceladus has already been investigated (Ref. 49 in the manuscript), and no P₃M was detected. The authors should therefore first clarify what is meant by “space” in this context. Second, if the authors indeed anticipate the presence of P₃M in open space, they should explain how this expectation is reconciled with the previously reported non-detection. For example, do they suggest that the concentration of P₃M was below the detection limit of that mission, such that more sensitive future measurements might reveal its presence?

Response:

Thanks a lot for your constructive suggestion. According to your suggestion, we have added some supplementary explanations and discussion at the corresponding positions in the main text (Lines 466-476), which are marked in blue and expressed as follows:

“Recent research has reported that heat flows can solubilize apatite, subsequently producing P₃m and then enhancing phosphate availability for prebiotic chemistry⁴⁸. In 2023, Frank Postberg et al reported the detection of huge phosphates originating from Enceladus’s ocean by Cassini’s Cosmic Dust Analyzer (CDA)⁴⁹. In addition, the suitable temperature⁵⁰ and similar heat flow system⁵¹ with the above research report have been found and confirmed in the Enceladus plume. Therefore, we speculate that P₃m may exist across space, such as in Enceladus’s ocean. Due to the high reactivity of P₃m in an alkaline aqueous environment, P₃m has difficulty surviving in alkaline aqueous solutions unless it is in an anhydrous solid state. It is prone to hydrolyze into orthophosphate, leading to a

concentration too low to be detectable. That may be one of the reasons why P_{3m} has not been detected in that mission by CDA.”

Reviewer 3

The authors have responded adequately to all of my criticisms. Publication is now recommended. I suggest that the authors include somewhere in the manuscript or SI the discussion provided in the letter of the measured doses and how it leads to anomalously high *G* values.

Response:

Thank you very much for your great suggestion. According to your suggestion, the content about the introduction of measured radiation doses have been supplemented into the Method Section in the main text (Lines 520-527), which are marked in blue and expressed as follows:

“The total radiation dose received by each unit was monitored from predeployed lithium 6 or 7 passive detection chips in the sample unit, including the doses received while standby time inside the cabin. All the values presented in this work are based on the calculated average dose to ensure a consistent frame of reference. The radiation absorbed in this experiment unit mainly consisted of radiation with LET values less than 10 keV/μm, including protons, neutrons and some heavy ion radiation. More detailed information is illustrated in the Supplementary Information, including the estimation of the radiation yield (*G* value).”

At the same time, the content about *G* values discussion have been supplemented into the SI of our manuscript (Chapter 22 of the Supplementary Information).